# ON THE MARGINAL REGRET BOUND MINIMIZATION OF ADAPTIVE METHODS

## ABSTRACT

Numerous adaptive algorithms such as AMSGrad and Radam have been proposed and applied to deep learning recently. However, these modifications do not improve the convergence rate of adaptive algorithms and whether a better algorithm exists still remains an open question. In this work, we propose a new motivation for designing the proximal function of adaptive algorithms, named as *marginal regret bound minimization*. Based on such an idea, we propose a new class of adaptive algorithms that not only achieves marginal optimality, but can also potentially converge much faster than any existing adaptive algorithms in the long term. We show the superiority of the new class of adaptive algorithms both theoretically and empirically using experiments in deep learning.

## 1 INTRODUCTION

Accelerating the convergence speed of optimization algorithms is one main concern of the machine learning community. After stochastic gradient descent (SGD) was introduced, quite a few variants of SGD have become popular, such as momentum (Polyak, 1964) and AdaGrad (Duchi et al., 2011). Instead of directly moving parameters in the negative direction of the gradient, AdaGrad proposed to scale the gradient by a matrix, which was the matrix in the proximal function of the composite mirror descent rule (Duchi et al., 2011). The diagonal version of AdaGrad designed this matrix to be the square root of the global average of the squared gradients. Duchi et al. (2011) proved that this algorithm could be faster than SGD when the gradients were sparse.

However, AdaGrad's performance is known to deteriorate when the gradients are dense, especially in high dimensional problems such as deep learning (Reddi et al., 2018). To tackle this issue, many new algorithms were proposed to boost the performances of AdaGrad. Most of these algorithms focused on changing the design of the matrix in the proximal function. For example, RMSProp (Tieleman & Hinton, 2012) and Adam (Kingma & Ba, 2015) changed the global average design in AdaGrad to the exponential moving average. However, Reddi et al. (2018) proved that such a modification had convergence issues in the presence of high frequency noises and added a max operation to the matrix of Adam, leading to the AMSGrad algorithm. Other modifications, such as Padam (Chen & Gu, 2018), AdaShift (Zhou et al., 2019), NosAdam (Huang et al., 2019), and Radam (Liu et al., 2019), were based on various designs of this matrix as well. However, all aforementioned works did not improve the convergence rate of AdaGrad and simply supported their designs using experiments and synthetic examples. A theoretical foundation for the design of this matrix that improves the convergence and guides future adaptive algorithms is very much needed.

In this work, we bring new insights to the design of the matrix in the proximal function. In particular, our major contributions in this paper are listed as follows

- We propose a new motivation for designing the proximal function in adaptive algorithms. Specifically, we have found a marginally optimal design, which is the best matrix at each time step through minimizing the marginal increment of the regret bound.

- Based on our proposal of *marginal regret bound minimization*, we create a new class of adaptive algorithms, named as AMX. We prove theoretically that AMX can converge with a regret bound of size $\tilde{O}(\sqrt{\tau})$, where $\tau$ is smaller than $T$. Such a regret bound is potentially much smaller than those of common adaptive algorithms and can make AMX converge

much faster than any existing adaptive algorithms, depending on $\tau$. In the worst case, we show it is at least as fast as AMSGrad and AdaGrad under the same assumptions

- We evaluate AMX's empirical performance on different tasks in deep learning. All experiments show our algorithm can converge fast and achieve good testing performances.

## 2 BACKGROUND

**Notation:** We denote the set of all positive definite matrices in $\mathbb{R}^{d \times d}$ by $\mathcal{S}_d^+$. For any two vectors $a, b \in \mathbb{R}^d$, we use $\sqrt{a}$ for element-wise square root, $a^2$ for element-wise square, $|a|$ for element-wise absolute value, $a/b$ for element-wise division, and $\max(a, b)$ for element-wise maximum between $a$ and $b$. We also frequently use the notation $g_{1:T,i} = [g_{1,i}, g_{2,i}, \cdots, g_{T,i}]$, i.e. the vector of all the $i$-th elements of vectors $g_1, g_2, \cdots g_T$. For a vector $a$, we use $\text{diag}(a)$ to represent the diagonal matrix whose diagonal entries are $a$. For two functions $f(t), g(t), f(t) = o(g(t))$ means $f(t)/g(t) \to 0$ as $t$ goes to infinity. We use $\tilde{O}(\cdot)$ to omit logarithm terms in big-$O$ notations. We say a space $\mathcal{X}$ has a bounded diameter $D_\infty$ if $\|x - y\|_\infty \le D_\infty, \forall x, y \in \mathcal{X}$.

**Online Learning Framework**. We choose the online learning framework to analyze all the algorithms in this paper. In this framework, an algorithm picks a new $x_t \in \mathcal{X}$ according to its update rule at each iteration $t$, where $X \subseteq \mathbb{R}^d$ is the set of feasible values of $x_t$. The composite loss function $f_t + \phi$ is then revealed, where $\phi$ is the regularization function that controls the complexity of $x$ and $f_t$ can be considered as the instantaneous loss at $t$. In the convex setting, $f_t$ and $\phi$ are both convex functions. The regularized regret function is defined with respect to an optimal predictor $x^*$ as

$$R(T) = \sum_{t=1}^T f_t(x_t) - f_t(x^*) + \phi(x_t) - \phi(x^*).$$

Our goal is to find algorithms that ensures a sub-linear regret, i.e. $R(T) = o(T)$, which means that the average regret converges to zero. For example, online gradient descent is proved to have a regret of $O(\sqrt{dT})$ (Zinkevich, 2003), where $d$ is the dimension size of $\mathcal{X}$. Note that stochastic optimization and online learning are basically interchangeable (Cesa-Bianchi et al., 2004). Therefore, we will refer to online algorithms and their stochastic counterparts using the same names. For example, we will use stochastic gradient descent (SGD) to represent online gradient descent as it is more well-known.

**Composite Mirror Descent Setup**. In this paper, we will revisit the general composite mirror descent method (Duchi et al., 2010b) used in the creation of the first adaptive algorithm, AdaGrad, to bring new insights into adaptive methods. Such a general framework is preferred because it covers a wide range of algorithms, including both SGD and all the adaptive methods, and thus simplifies the discussions. The composite mirror descent rule at the time step $t + 1$ is to solve for

$$x_{t+1} = \text{argmin}_{x \in \mathcal{X}}\{\alpha_t \langle g_t, x \rangle + \alpha_t \phi(x) + B_{\psi_t}(x, x_t)\}, \tag{1}$$

where $g_t$ is the gradient, $\phi(x)$ is the regularization function in the dual space, and $\alpha_t$ is the step size. Also, $\psi_t$ is a strongly convex and differentiable function, named as the *proximal function* and $B_{\psi_t}(x, x_t)$ is the Bregman divergence associated with $\psi_t$ defined as

$$B_{\psi_t}(x, y) = \psi_t(x) - \psi_t(y) - \langle \nabla\psi_t(y), x - y \rangle.$$

The general update rule (1) is mostly determined by the function $\psi_t$. We first observe that it becomes the projected SGD algorithm when $\psi_t(x) = x^T x$ and $\phi(x) = 0$.

$$x_{t+1} = \text{argmin}_{x \in \mathcal{X}}\{\alpha_t \langle g_t, x \rangle + \|x - x_t\|_2^2\} = \Pi_{\mathcal{X}}(x_t - \alpha_t g_t), \tag{SGD}$$

where $\Pi_{\mathcal{X}}(x) = \text{argmin}_{y \in \mathcal{X}}\|x - y\|_2$ is the projection operation that ensures the updated parameter is in the original space. On the other hand, adaptive algorithms choose different proximal functions $\psi_t(x) = \langle x, H_t x \rangle$, where $H_t$ can be any full or diagonal symmetric positive definite matrix.

$$x_{t+1} = \text{argmin}_{x \in \mathcal{X}}\{\alpha_t \langle g_t, x \rangle + \alpha_t \phi(x) + \langle x - x_t, H_t(x - x_t) \rangle\}, \tag{Adaptive}$$

Another popular representation of adaptive algorithms is the generalized projection rule $x_{t+1} = \Pi_{\mathcal{X},H_t}(x_t - \alpha_t H_t^{-1} g_t)$, where $\Pi_{\mathcal{X},H_t}(x) = \text{argmin}_{y \in \mathcal{X}}\|H_t^{1/2}(x - y)\|_2$, which is used in a lot of

recent literature such as Reddi et al. (2018); Huang et al. (2019). We show that these two rules are actually equivalent when $\phi(x) = 0$ in the Appendix A.1, so that the regret bounds found in different representations can be generalized. A few recent works have shown that adaptive algorithms work well with special designs of the step sizes $\alpha_t$ (Choi et al., 2019; Vaswani et al., 2020). In this work, we choose the more standard $\alpha_t = \alpha/\sqrt{t}$ as it is used in most analysis. Also, we restrain our discussions to diagonal matrix proximal functions in the main text, i.e. $H_t = \text{diag}(h_t), h_t \in \mathbb{R}^d$. Discussions on extending our results to full matrix proximal functions are provided in Appendix B.

**Different Designs of the Proximal Function.** Recently, researchers have proposed numerous designs of $H_t = \text{diag}(h_t)$, such as AdaGrad (Duchi et al., 2011), Adam (Kingma & Ba, 2015), AMSGrad (Reddi et al., 2018) and NosAdam (Huang et al., 2019), to name a few. It's impossible to go over all the proposed designs in this section so we choose the two most famous designs to review. The first adaptive algorithm, AdaGrad, used the square root of the average of past gradient squares as the diagonal $h_t$ of the matrix in the proximal function (Duchi et al., 2011), i.e.

$$h_t = \left(\frac{\sum_{i=1}^t g_i^2}{t}\right)^{1/2} \tag{AdaGrad}$$

Normally, a small constant $\epsilon$ is added to $h_t$ at each iteration. Some recent work have shown that tuning this constant can benefit the performance of adaptive algorithms (Zaheer et al., 2018; Savarese et al., 2019). However, in this work, we assume it is small and fixed for simplicity, as it is originally designed to compute the pseudo-inverse, or equivalently, avoid division by zero in the generalized projected descent. Kingma & Ba (2015) proposed Adam to replace the simple average by exponential moving average in AdaGrad, but Reddi et al. (2018) showed that there was a mistake in Adam's convergence analysis, which lead to divergence of Adam even in simple convex problems. They therefore proposed the following simple modification of Adam to ensure its convergence.

$$h_t = \max_t\left\{\frac{\sum_{i=1}^t (1 - \beta_2)\beta_2^{t-i} g_i^2}{t}\right)^{1/2}\right\} \tag{AMSGrad}$$

where $\beta_2 \in (0, 1)$ is a constant. We propose the following theorem that generalizes the regret bounds for most of the designs of the diagonal matrix proximal function.

**Theorem 2.1** *Let the sequence $\{x_t\}$ be defined by the update rule (1) and for any $x^*$, denote $D_{t,\infty}^2 = \|x_t - x^*\|_\infty^2$. When $\psi_t(x) = \langle x, H_t x\rangle$, where $H_t = \text{diag}(h_{t,1}, h_{t,2}, \cdots, h_{t,d})$, assume without loss of generality that $\phi(x_1) = 0, H_0 = 0$, if $(h_{t,i}/\alpha_t) \geq (h_{t-1,i}/\alpha_{t-1})$, then*

$$R(T) \leq \sum_{t=1}^T \sum_{i=1}^d \left(\frac{h_{t,i}}{\alpha_t} - \frac{h_{t-1,i}}{\alpha_{t-1}}\right)D_{t,\infty}^2 + \sum_{t=1}^T \sum_{i=1}^d \frac{\alpha_t g_{t,i}^2}{2h_{t,i}}. \tag{2}$$

The proof is relegated to Appendix A.3. The above regret bound is suitable for any designs of $h_t$ that satisfy the constraint condition $(h_{t,i}/\alpha_t) \geq (h_{t-1,i}/\alpha_{t-1})$. Such a condition is crucial because if it is unsatisfied, the regret $R(T)$ might diverge. In fact, the divergence of Adam in simple convex problems results from not satisfying this constraint, which is proved by Reddi et al. (2018). With $\alpha_t = \alpha/\sqrt{t}$ in Theorem 2.1, most of recent adaptive algorithms have a regret bound of the following form (Duchi et al., 2011; Reddi et al., 2018; Huang et al., 2019; Luo et al., 2019).

$$R(T) \leq C_1\sqrt{T}\sum_{i=1}^d h_{T,i} + f(T)\sum_{i=1}^d \|g_{1:T,i}\|_2 + C_2 \tag{3}$$

where $C_1, C_2$ are constants and $f(T) = o(\sqrt{T})$. [1] These algorithms are supposed to converge faster than SGD when the gradients are sparse or small, i.e. when $\sum_{i=1}^d h_{T,i} \ll \sqrt{d}$ and $\sum_{i=1}^d \|g_{1:T,i}\|_2 \ll \sqrt{dT}$. However, all existing regret bounds are still $O(\sqrt{T})$, which makes it hard to compare different proximal functions. Whether the best proximal function exists and whether the $O(\sqrt{T})$ regret bound can be further improved still remain open questions.

---

[1] The regret bound of AdaGrad given in the original paper was $O(\|g_{1:T,i}\|_2)$ because Duchi et al. (2011) used a constant learning rate $\alpha_t = \alpha$ and hence $h_t = \sum_{i=1}^t g_i^2$. When changed into the same setting where $\alpha_t = \alpha/\sqrt{t}$, $h_t$ becomes $\sum_{i=1}^t g_i^2/t$ and the regret also has the form of (3), shown in Reddi et al. (2018).

## 3   THE MOTIVATION-MARGINAL REGRET BOUND MINIMIZATION

In this section, we introduce the motivation behind our new class of algorithms. Although we find it difficult to determine the optimal proximal function globally, we show that it is possible to find the best proximal function at each iteration through *marginal regret bound minimization*. Denote $\tilde{R}(T)$ to be the regret upper bound (the right hand side of inequality (2)) in Theorem 2.1. At time step $T$, we define the marginal regret bound increment $\Delta\tilde{R}(T)$ as follows.

$$\Delta\tilde{R}(T) := \tilde{R}(T) - \tilde{R}(T-1) = \sum_{i=1}^{d}(\frac{h_{T,i}}{\alpha_T} - \frac{h_{T-1,i}}{\alpha_{T-1}})D_{T,\infty}^2 + \sum_{i=1}^{d}\frac{\alpha_T g_{T,i}^2}{2h_{T,i}}.$$

As shown in the definition, $\Delta\tilde{R}(T)$ will be the increment in the regret bound $\tilde{R}(T)$ after $h_T$ is determined. An important observation here is that $h_{T-1}$ is a given constant at $T$, so $\Delta\tilde{R}(T)$ is only a function of $h_T$. Therefore, the best design of $h_T$ we can find at this moment is the one that minimizes $\Delta\tilde{R}(T)$ and satisfy the constraint in Theorem 2.1. Consider the minimization problem

$$\min_{h_T}\Delta\tilde{R}(T), \text{ s.t.} \frac{h_{T,i}}{\alpha_T} \geq \frac{h_{T-1,i}}{\alpha_{T-1}} \geq 0, \tag{4}$$

We propose the following proposition that solves the problem above

**Proposition 3.1**  *With $\alpha_t = \alpha/\sqrt{t}$, the minimum of problem (4) is obtained at*

$$h_T^* = max\left\{\sqrt{\frac{T-1}{T}}h_{T-1}, (\frac{\alpha^2}{2TD_{T,\infty}^2})^{1/2}|g_T|\right\}. \tag{5}$$

To see this, set the function $L$ with the Lagrangian multiplier $\mu$ as follows

$$L(h,\alpha) = \sum_{i=1}^{d}(\frac{h_{T,i}}{\alpha_T} - \frac{h_{T-1,i}}{\alpha_{T-1}})D_{T,\infty}^2 + \sum_{i=1}^{d}\frac{\alpha_T g_{T,i}^2}{2h_{T,i}} - \langle\frac{h_T}{\alpha_T} - \frac{h_{T-1}}{\alpha_{T-1}}, \mu\rangle.$$

Take partial derivatives with respect to each $h_{T,i}$, we can see that $D_{T,\infty}^2 = (\alpha_T^2 g_{T,i}^2)/2h_{T,i}^2 + \mu_i$. By the complementary slackness conditions, either $\mu_i = 0$ or $h_{T,i} = (\alpha_T/\alpha_{T-1})h_{T-1,i}$. When $\mu_i = 0$, $h_{T,i} = (\alpha_T^2/2D_{T,\infty}^2)^{1/2}|g_T|$ and the constraint condition $h_{T,i} \geq (\alpha_T/\alpha_{T-1})h_{T-1,i}$ needs to be satisfied. Hence, by setting $\alpha_T = \alpha/\sqrt{T}$, we can get the solution in (5).

Solution (5) is the best diagonal matrix proximal function in terms of regret bound increment at time $T$. Therefore, if we replace $T$ by $t$ in the subscripts, we can obtain a greedy choice of the proximal function $h_t$ that minimizes the marginal regret bound increment at each time step. Intuitively, the reason this solution achieves the minimum is that it balances the two terms of $\Delta\tilde{R}(T)$. On each dimension $(i)$, it makes the first term of $\Delta\tilde{R}(T)$ zero when the derivative is small [2], i.e. when we don't need to have a even larger $h_{T,i}$ to slow down. When the derivative is too large, $h_{T,i}$ adapts to the size of the derivative so that the second term of $\Delta\tilde{R}(T)$ is constrained.

However, similar to the other greedy algorithms, solution (5) is only suboptimal as it minimizes the regret bound marginally instead of globally. Moreover, the parameter $D_{t,\infty}$ is often unknown during the optimization process because $x^*$ is usually unknown. Therefore, stronger theoretical motivation is needed to trust that solution (5) or similar algorithms can be useful and beneficial.

## 4   A NEW CLASS OF ADAPTIVE ALGORITHMS - AMX

Now, motivated by the greedy choice of $h_t$ in section 3, we focus on a more general design of the diagonal matrix in the proximal functions that has the following form and show why such greedy designs can be beneficial in the long term. Consider

$$h_t = \max\left\{\sqrt{\frac{t-1}{t}}h_{t-1}, c_t|g_t|\right\}. \tag{AMX}$$

---

[2]Because $(h_{t,i}/\alpha_t) - (h_{t-1,i}/\alpha_{t-1}) = 0$ when $h_{t,i} = (\alpha_t/\alpha_{t-1})h_{t-1,i}$

---

**Algorithm 1** AMX Algorithm (Diagonal, Composite Mirror Descent Form)

---

1: **Input:** $x \in \mathcal{F}$, $\alpha_t = \alpha/\sqrt{t}$, $\{c_t\}_{t=1}^T$, $\phi(x)$, $\epsilon$
2: **Initialize** $h_0 = 0$, $H_0 = 0$
3: **for** $t = 1$ **to** $T$ **do**
4: $\quad g_t = \nabla f_t(x_t)$
5: $\quad h_t = \max(\sqrt{\frac{t-1}{t}} h_{t-1}, c_t|g_t|) + \epsilon$
6: $\quad x_{t+1} = \operatorname{argmin}_{x \in \mathcal{X}} \{\alpha_t \langle g_t, x \rangle + \alpha_t \phi(x) + \langle x - x_t, \operatorname{diag}(h_t)(x - x_t) \rangle \}$
7: **end for**

---

where $c_t$ is an arbitary function of $t$, for example solution (5) or as simple as $c_t = 1$. The corresponding new class of adaptive algorithms is given in Algorithm 1, which we name as AMX. Note that the diagonal proximal function performs all operations coordinate-wisely, therefore we start our analysis from one dimension $(i)$. Denote $(i)$-th dimension component of the regret bound as

$$\tilde{R}^{(i)}(T) := \sum_{t=1}^T (\frac{h_{t,i}}{\alpha_t} - \frac{h_{t-1,i}}{\alpha_{t-1}}) D_{t,\infty}^2 + \sum_{t=1}^T \frac{\alpha_t g_{t,i}^2}{2h_{t,i}}. \tag{6}$$

For a sequence of gradients $g_1, g_2, \cdots, g_T$, denote the time steps $t$ when $h_{t,i} = c_t|g_{t,i}|$ as $\tau_1^{(i)}, \tau_2^{(i)}, \cdots \tau_{m_i}^{(i)}$. Note that $\tau_j^{(i)}$'s may be different across different dimensions, so the analysis apply to each dimension independently. These are the time steps when the gradient term $c_t|g_t|$ dominates $h_t$ on the $(i)$-th dimension and $h_{t,i} = c_t|g_{t,i}|$ when $t = \tau_j^{(i)}, \forall j = 1, \cdots, m_i$. Since $h_{t,i}$ is equal to $\sqrt{\frac{t-1}{t}} h_{t-1,i}$ between $\tau_j^{(i)}$'s, which is exactly the same as in section 3, the increment in the first term of the right hand side of (6) is always 0. Therefore, we have the following proposition

**Proposition 4.1** *For any $\tau \in (\tau_j^{(i)}, \tau_{j+1}^{(i)})$, the regret bound increment on the $(i)$-th dimension is*

$$\tilde{R}^{(i)}(\tau) - \tilde{R}^{(i)}(\tau_j^{(i)}) = \sum_{t=\tau_j^{(i)}+1}^\tau \frac{\alpha_t g_{t,i}^2}{2h_{t,i}}.$$

The above proposition indicates that the regret increments between the $\tau_j^{(i)}$'s are only related to the second term of $\tilde{R}^{(i)}(T)$. Note that the first term of $\tilde{R}^{(i)}(T)$ is a major reason why the regret bound is $O(\sqrt{T})$ because there is a $1/\alpha_T$ term in the summation. Therefore, such designs of $h_{t,i}$ try to constrain the regret increments between $\tau_j^{(i)}$'s and hence can potentially make the regret bound small. More importantly, Proposition 4.1 is true for the time steps between the last $\tau_{m_i}^{(i)}$ and $T + 1$. Denote $D_\infty$ to be the bounded diameter of the parameter space $\mathcal{X}$, because the regret bound increment is only related to the second term after the last $\tau_{m_i}^{(i)}$, the bound in equation (6) becomes

$$\tilde{R}^{(i)}(T) \leq \sum_{t=1}^T (\frac{h_{t,i}}{\alpha_t} - \frac{h_{t-1,i}}{\alpha_{t-1}}) D_\infty^2 + \sum_{t=1}^T \frac{\alpha_t g_{t,i}^2}{2h_{t,i}} \leq \frac{D_\infty^2 \sqrt{\tau_{m_i}^{(i)}}}{\alpha} h_{\tau_{m_i}^{(i)},i} + \sum_{t=1}^T \frac{\alpha}{2\sqrt{t}c_t} |g_{t,i}| \tag{7}$$

Since $\tilde{R}(T) = \sum_{i=1}^d \tilde{R}^{(i)}(T)$, so the total regret upper bound across all the dimensions is

$$R(T) \leq \sum_{i=1}^d \tilde{R}^{(i)}(T) \leq \sum_{i=1}^d \frac{D_\infty^2 \sqrt{\tau_{m_i}^{(i)}}}{\alpha} h_{\tau_{m_i}^{(i)},i} + \sum_{i=1}^d \sum_{t=1}^T \frac{\alpha}{2\sqrt{t}c_t} |g_{t,i}| \tag{8}$$

The first term of the right hand side can be considered as better than the $O(\sqrt{T})$ regret bound of SGD or common adaptive algorithms when some or all of the $\tau_{m_i}^{(i)}$ are much smaller than $T$ and $h_{t,i}$'s are bounded. Therefore if we can ensure the second term also increases much slower than $O(\sqrt{T})$, which is decided by the design of $c_t$, then the AMX class of algorithms is potentially much faster than SGD and the other adaptive algorithms. Note that we need $h_{t,i}$ to be bounded in the above arguments, therefore $c_t$ can be at most a constant. Fortunately, one simple yet very effective design that we have found is $c_t = 1$. We formalize the above statements in the following theorem.

**Theorem 4.1** *Let $\{x_t\}$ and $\{h_t\}$ be the sequences obtained from Algorithm 1, $\alpha_t = \alpha/\sqrt{t}, c_t = 1$. Let $\{\tau_{m_i}^{(i)}\}_{i=1}^d$ be the largest time steps $t$ on each dimension when $h_{t,i} = c_t|g_{t,i}|$. Assume that $\mathcal{F}$ has bounded diameter $\|x - y\|_\infty \le D_\infty, \forall x, y \in \mathcal{F}$. Then we have the following bound on the regret.*

$$R(T) \le \sum_{i=1}^d \frac{D_\infty^2 \sqrt{\tau_{m_i}^{(i)}}}{\alpha} h_{\tau_{m_i}^{(i)},i} + \frac{\alpha}{2} \sum_{i=1}^d \sqrt{1 + \log \tau_{m_i}^{(i)}} \|g_{1:\tau_{m_i}^{(i)},i}\|_2 + \frac{\alpha}{2} \sum_{i=1}^d \sqrt{\tau_{m_i}^{(i)}} \log(\frac{T}{\tau_{m_i}^{(i)}})|g_{\tau_{m_i}^{(i)},i}|, \tag{9}$$

An important remark here is that using a different constant $c_t = c$ in Theorem 4.1 is equivalent to tuning the step size by $1/c$, as it magnifies all $|g_t|$ at the same time in the algorithm. Using a decaying $c_t$ can further improve the first term in Theorem 4.1, but it also enlarges the second and the third term so the regret may not be $O(\sqrt{T})$ anymore. We will focus on $c_t = 1$ to prove AMX can potentially converge faster in the rest of this paper, but a detailed discussion about the possible choices of $c_t$ for future designs of the AMX algorithm is provided in Appendix A.5, which proves that $c_t$ cannot be $O(1/\sqrt{t})$ if we do not impose any further assumptions .

Now, since most of the bound in Theorem 4.1 depends on the time steps $\{\tau_{m_i}^{(i)}\}_{i=1}^d$ instead of $T$ (only a log term), the specific AMX algorithm can be potentially much faster than common adaptive algorithms. To further make our argument clearer, we propose the following corollary.

**Corollary 4.1** *Let $\tau = max_i\{\tau_{m_i}^{(i)}\}$ in Theorem 4.1, under the same assumptions as AdaGrad and AMSGrad, Algorithm 1 converges with regret bound*

$$O(max(\sqrt{\tau}, \sqrt{\tau} \log \frac{T}{\tau})) \tag{10}$$

The corollary indicates that the regret bound is approximately of size $\tilde{O}(\sqrt{\tau})$ if we omit the log terms. As far as we are aware, this is the first algorithm that generates a regret bound that is not asymptotically $O(\sqrt{T})$, so AMX can be potentially much faster than any existing algorithms. The time step $\tau$ can be understood as "the time when the gradients start to converge" and whether it makes the convergence faster depends on the distribution of the gradients. For example, if $\tau = \sqrt{T}$, the regret bound is only of size $O(T^{1/4} \log T)$. We emphasize that a small $\tau$ is not an assumption on the gradient distribution, but rather a condition that once satisfied, the regret bound will only increase logarithmically and hence the algorithm converges very fast. Moreover, the regret bounds of the other adaptive algorithms go to $O(\sqrt{T})$ even under such conditions, because their regret bound increments are not minimized. We use a rather simple example to illustrate why AMX has this unique advantage.

**Example.** Suppose that the domain is a hyper-cube $\mathcal{X} = \{\|x\|_\infty \le 1\}$, then $D_\infty = 2$. Assume that on each dimension, the gradient decreases as $|g_{t,i}| = (1/\sqrt{t})|g_{1,i}|$, and $|g_{1,i}| \ll 1, \forall i$. Note that this is one example where adaptive algorithms should work well since $\|g_{1:T,i}\|_2 \le |g_{1,i}|\sqrt{(1 + \log T)} \ll \sqrt{T}$. A very important property for AMX in this case is that $\tau$ is the first time step, so its regret bound only increases logarithmically. However, the regret bounds of the other algorithms still goes to $O(\sqrt{T})$. We plot the regret bounds of AMX, AMSGrad and AdaGrad in Figure 4. Note that the regret bound of AMX increases much slower than AdaGrad and AMSGrad, hence it is much faster than these algorithms in this example. One may argue that the example is extreme since $\tau = 1$ rarely happens in real situations. However, the regret in this example can be understood as the regret increment after $\tau$ in real training processes, i.e. before $\tau$, AMX is only asymptotically as fast as the other adaptive algorithms, but after $\tau$, since the regret increment of AMX is very small, it converges very fast. More details of this example can be found in Appendix D.1.

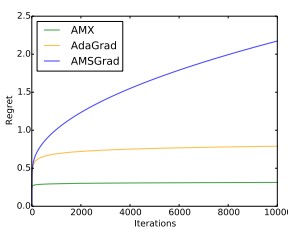

Figure 1: The regret bounds of AMX, AdaGrad, AMSGrad in the example.

Besides, since the term $\sqrt{\tau} \log(T/\tau)$ in Corollary 4.1 is at most $O(\sqrt{T})$[3], the AMX algorithm is at least as fast as AdaGrad and AMSGrad under the same assumptions. We propose the following theorem that corresponds to the general results in section 2 to prove our claim:

---

[3]This claim can be easily proved by taking derivative of $\tau$ and finding the maximum of $\sqrt{\tau} \log(T/\tau)$

**Theorem 4.2** *Let $\{x_t\}$ and $\{h_t\}$ be the sequences obtained from Algorithm 1, $\alpha_t = \alpha/\sqrt{t}, c_t = 1$. Assume that $\mathcal{F}$ has bounded diameter $\|x - y\|_\infty \leq D_\infty, \forall x, y \in \mathcal{F}$. Then we have the following bound on the regret.*

$$R(T) \leq \frac{D_\infty^2 \sqrt{T}}{\alpha} \sum_{i=1}^{d} h_{T,i} + \frac{\alpha}{2}\sqrt{1 + \log T} \sum_{i=1}^{d} \|g_{1:T,i}\|_2, \quad (11)$$

The above bound can be considered as being better than the regret of SGD, i.e., $O(\sqrt{dT})$, when $\sum_{i=1}^{d} h_{T,i} \ll \sqrt{d}$ and $\sum_{i=1}^{d} \|g_{1:T,i}\|_2 \ll \sqrt{dT}$ (Duchi et al., 2011). Therefore, AMX can be at least much faster than SGD when the gradients are sparse or small, and it can be potentially even faster. To keep up with the current popular adaptive algorithms such as Adam, we also provide the detailed implementation of adding first order momentum into AMX and include some discussions on its convergence properties in Appendix C. Similar to Algorithm 1, the AMX with momentum algorithm has a regret bound that (mostly) depends on $\tau$ instead of $T$ and hence enjoys the acceleration.

## 5 EXPERIMENTS

In this section, we evaluate the effectiveness of the specific AMX algorithm proposed in Section 4 (i.e. $c_t = 1$) on different deep learning tasks. We relegate more details of parameter tuning and step size decay strategies to Appendix D.2-D.5. Moreover, an empirical analysis for different designs of $\{c_t\}_{t=1}^{T}$ that show different $\{c_t\}_{t=1}^{T}$'s generate different performances is provided in Appendix D.6.

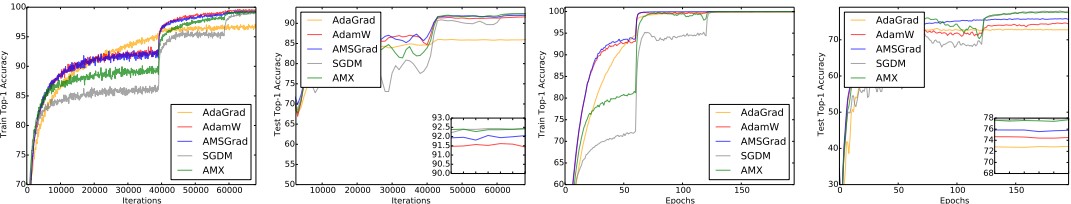

(a) CIFAR-10 Training Acc.(b) CIFAR-10 Testing Acc(c) CIFAR-100 Training Acc(d) CIFAR-100 Testing Acc.

Figure 2: Training and Testing Top-1 accuracy on CIFAR-10 and CIFAR-100.

Table 1: Testing Top-1 accuracy on the CIFAR-10, CIFAR-100 datasets and testing IoU on the VOC2012 Segmentation dataset. The results were averaged over 5 independent runs. Our results were shown in bold.

| OPTIMIZER | CIFAR-10 | CIFAR-100 | VOC2012 |
|---|---|---|---|
| SGDM | $92.40 \pm 0.06$ | $77.80 \pm 0.08$ | $76.10 \pm 0.09$ |
| ADAGRAD | $85.60 \pm 0.14$ | $72.84 \pm 0.06$ | $71.28 \pm 0.18$ |
| ADAM | $91.85 \pm 0.03$ | $74.51 \pm 0.05$ | $73.32 \pm 0.21$ |
| AMSGRAD | $91.97 \pm 0.07$ | $75.75 \pm 0.04$ | $73.66 \pm 0.13$ |
| **AMX** | $\mathbf{92.42 \pm 0.08}$ | $\mathbf{77.65 \pm 0.10}$ | $\mathbf{76.04 \pm 0.16}$ |

Table 2: Validation perplexity on the character Penn Tree Bank (PTB) dataset and BLEU score on the IWSLT'14 DE-EN dataset. The results were averaged over three independent runs. Our results were shown in bold.

| OPTIMIZER | CHAR-PTB | IWSLT'14 |
|---|---|---|
| ADAGRAD | $2.63 \pm 0.04$ | $25.56 \pm 0.05$ |
| ADAM | $2.48 \pm 0.08$ | $28.01 \pm 0.07$ |
| AMSGRAD | $2.46 \pm 0.05$ | $28.15 \pm 0.06$ |
| **AMX** | $\mathbf{2.32 \pm 0.04}$ | $\mathbf{28.29 \pm 0.03}$ |

We compared our AMX algorithm with SGD with momentum (SGDM), Adam, AdaGrad and AMSGrad on different tasks in deep learning. The hyper-parameters in AMX were set to be $c_t = 1$ in this subsection. For the language modeling and the neural machine translation tasks, because SGDM typically performs much worse than adaptive algorithms, we did not include it in the comparisons. Following Loshchilov & Hutter (2019), we used decoupled weight decay in all the algorithms.

**Image Classification.** We first conducted some experiments where $\tau$ was possibly very large and AMX was only as fast as the other adaptive algorithms, but it still achieved better testing performances. The image classification task was performed on the CIFAR (Krizhevsky et al., 2009) datasets. We used the publicly available code by Li et al. (2020) and DeVries & Taylor (2017) to train ResNet-20 and ResNet-18 (He et al., 2016) on CIFAR-10 and CIFAR-100 respectively using batch size of 128.

We summarized the performances of different algorithms in Figure 2 and Table 1. As observed, AMX started slightly slowly, but it quickly caught up with the other adaptive algorithms and converged much faster than SGDM. This was possibly because the time when gradients start to converge (the $\tau$ in section 4) was large in image classification tasks, and AMX could only converge asymptotically as fast as the other adaptive algorithms, corresponding to Theorem 4.1. However, its final testing performance was as good as SGDM, so it converged both fast and well at the same time. The other adaptive algorithms such as Adam and AMSGrad had faster training performances in the beginning, but they ended up with much worse final accuracy than SGDM and AMX.

**Image Segmentation.** Next, more experiments proved our claim that AMX could be potentially much faster and generate even better testing performances. For the segmentation task, we used the publicly released implementation of the Deeplab model (Chen et al., 2016) by Kazuto1011 (2016) and evaluated the performances of different algorithms on the PASCAL VOC2012 Segmentation dataset (Everingham et al., 2014). We used a small batch size of 4 and a polynomially decaying step size in 20k iterations. The trained models were evaluated at the 5k, 10k, 15k and 20k iterations and we used mean Intersection over Union (IoU) as the evaluation metric. The results were provided in Figure 3(a), 3(b) and Table 1. As shown in the figures and the table, AMX was not only the fastest adaptive algorithm but also achieved the best IoU score, which was comparable to that of SGDM. The other algorithms were not able to perform similarly.

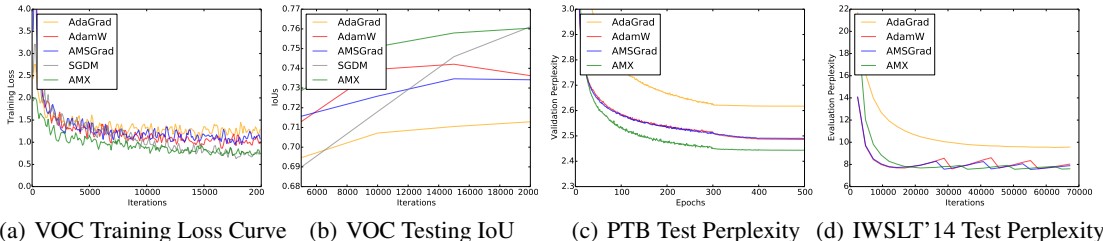

| (a) VOC Training Loss Curve | (b) VOC Testing IoU | (c) PTB Test Perplexity | (d) IWSLT'14 Test Perplexity |

Figure 3: (a), (b). Training Loss and Testing IoU curves on the VOC2012 Segmentation dataset. (c). Validation perplexity curve on the Penn Tree Bank (PTB) dataset. (d). Validation Perplexity curve on the IWSLT'14 DE-EN machine translation dataset.

**Language Modeling.** We trained three-layer LSTMs (Hochreiter & Schmidhuber, 1997) on the character level Penn Tree Bank (PTB) dataset. The general setup in Merity et al. (2017) was adopted in our experiments. Specifically, we trained the model for 500 epochs with batch size 128. The validation perplexity curve and the final validation perplexity were shown in Figure 3(c) and Table 2. It can be observed that AMX was the fastest algorithm and achieved the lowest perplexity among all the adaptive algorithms, which proved our claim that AMX was potentially much faster.

**Neural Machine Translation.** We utilized the publicly released code by pcyin (2018) and trained the basic attentional neural machine translation models (Luong et al., 2015) on the IWSLT'14 DE-EN (Ranzato et al., 2015) dataset. We used 64 as the batch size and decreased the step size by 2 every 5 iterations. The validation perplexity curve and the final BLEU score were reported in Figure 3(d) and Table 2. AMX not only had a much smoother validation perplexity curve, but also achieved the best BLEU score among all the adaptive algorithms, showing that AMX was indeed a better choice.

## 6 CONCLUSION

In this paper, we propose our design of the best proximal functions at each time step based on *marginal regret bound minimization*. We then show that a more general class of adaptive algorithms can not only achieve marginal optimality in some sense, but also converge much faster than any existing adaptive algorithms, depending on the distribution of the gradients. We evaluate one particular case of our new class of algorithms on different tasks in deep learning and show its effectiveness. This work provides a new framework for adaptive algorithms and can hopefully prevent the random searching process for better designs of the proximal function. Future researchers can concentrate on finding better choices of the sequence $\{c_t\}_{t=1}^T$ to find better algorithms.

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

# A    PROOFS OF RESULTS IN THE MAIN TEXT

In this appendix, we provide proofs of all the results and theorems in the main text, except for Theorem 4.2, which is proved in section C as it is the same as $\beta_1 = 0$ in the momentum design. An important remark here is that adding $\epsilon$ to $h_t$ does not affect the proof here because when $h_{t,i}$ is placed on the denominator, $1/h_{t,i} \geq 1/(h_{t,i} + \epsilon)$. When $h_{t,i}$ is placed on the numerator, we generally use $h_{t,i}$ instead of giving it a specific form. Also, we set $\epsilon$ to be very small in real experiments (1e-8).

**Further Notations**. Except for the notations mentioned in the main text, the following notations are needed for this appendix. Given a norm $\|\cdot\|$, its dual norm is defined to be $\|y\|_* = \sup_{\|x\| \leq 1}\{\langle x, y \rangle\}$. For example, the dual norm of the Mahalanobis norm $\|\cdot\|_A = \sqrt{\langle \cdot, A \cdot \rangle}, A \succ 0$ is the Mahalabonis norm $\|\cdot\|_{A^{-1}}$. A function $f$ is said to be 1-strongly convex with respect to the norm $\|\cdot\|$ if

$$f(y) \geq f(x) + \langle \nabla f(x), y - x \rangle + \frac{1}{2}\|x - y\|$$

## A.1    EQUIVALENCE BETWEEN COMPOSITE MIRROR DESCENT AND GENERALIZED PROJECTED ALGORITHMS

Begining with the generalized projected adaptive algorithm, we can find that

$$
\begin{aligned}
x_{t+1} = \Pi_{\mathcal{X}}^{H_t}(x_t - \alpha_t H_t^{-1} g_t) &:= \operatorname{argmin}_{x \in \mathcal{X}} \|x - (x_t - \alpha_t H_t^{-1} g_t)\|_{H_t}^2 \\
&= \operatorname{argmin}_{x \in \mathcal{X}} \|H_t^{1/2}(x - (x_t - \alpha_t H_t^{-1} g_t))\|^2 \\
&= \operatorname{argmin}_{x \in \mathcal{X}} \|H_t^{1/2}(x - x_t) + \alpha_t H_t^{-1/2} g_t\|^2 \\
&= \operatorname{argmin}_{x \in \mathcal{X}} \{2\alpha_t \langle g_t, x - x_t \rangle + \langle x - x_t, H_t(x - x_t) \rangle\} \\
&= \operatorname{argmin}_{x \in \mathcal{X}} \{2\alpha_t \langle g_t, x \rangle + \langle x - x_t, H_t(x - x_t) \rangle\}
\end{aligned}
\tag{12}
$$

Given $\psi_t(x) = \langle x, H_t x \rangle$, it can be shown directly (also in A.3) that $B_{\psi_t}(x, x_t) = \langle x - x_t, H_t(x - x_t) \rangle$. Therefore comparing equation 12 with equation 1, we know that they are equivalent when the regularization term $\phi(x) = 0$, except that we need to double the step sizes $\alpha_t$.

## A.2    PROOF OF LEMMA A.1

**Lemma A.1** *Let $\{x_t\}$ be a sequence of outputs from the update rule (1) and assume that $\{\psi_t\}$ are strongly convex functions with respect to the norm $\|\cdot\|_{\psi_t}$. Let $\|\cdot\|_{\psi_t^*}$ be the corresponding dual norm. Also without loss of generality let $B_{\psi_0}(x^*, x_0) = 0$. Then for any $x^*$, we have*

$$
\sum_{t=1}^{T} f_t(x_t) - f_t(x^*) + \phi(x_t) - \phi(x^*) \leq \sum_{t=1}^{T} \frac{B_{\psi_t}(x^*, x_t)}{\alpha_t} - \frac{B_{\psi_{t-1}}(x^*, x_t)}{\alpha_{t-1}} + \frac{\alpha_t^2}{2}\|f_t'(x_t)\|_{\psi_t^*}^2.
\tag{13}
$$

**Proof**. Since $x_{t+1}$ satisfies equation (1), for all $x \in \mathcal{X}$ and $\phi'(x_{t+1}) \in \partial\phi(x_{t+1})$.

$$\langle x - x_{t+1}, \alpha_t g_t + \alpha_t \phi'(x_{t+1}) + \nabla \psi_t(x_{t+1}) - \nabla \psi_t(x_t) \rangle \geq 0 \tag{14}$$

Also since $f_t$'s are convex functions, we know that $f_t(x) \geq f_t(x_t) + \langle f'(x_t), x - x_t \rangle$ and likewise for $\phi(x_{t+1})$. We therefore have the following

$$
\begin{aligned}
&\alpha_t(f_t(x_t) - f_t(x^*)) + \alpha_t(\phi(x_{t+1}) - \phi(x^*)) \\
&\leq \alpha_t \langle f_t'(x_t), x_t - x^* \rangle + \alpha_t \langle \phi_t'(x_{t+1}), x_{t+1} - x^* \rangle \\
&= \alpha_t \langle f_t'(x_t), x_{t+1} - x^* \rangle + \alpha_t \langle \phi_t'(x_{t+1}), x_{t+1} - x^* \rangle + \alpha_t \langle f_t'(x_t), x_t - x_{t+1} \rangle \\
&= \langle x^* - x_{t+1}, \nabla \psi_t(x_{t+1}) - \nabla \psi_t(x_t) - \alpha_t f_t'(x_t) - \alpha_t \phi_t'(x_{t+1}) \rangle \\
&\quad + \langle x^* - x_{t+1}, \nabla \psi_t(x_t) - \nabla \psi_t(x_{t+1}) \rangle + \alpha_t \langle f_t'(x_t), x_t - x_{t+1} \rangle \\
&\leq \langle x^* - x_{t+1}, \nabla \psi_t(x_t) - \nabla \psi_t(x_{t+1}) \rangle + \alpha_t \langle f_t'(x_t), x_t - x_{t+1} \rangle
\end{aligned}
\tag{15}
$$

where the first inequality is due to the convexity of $\phi_t$ and $f_t$. The second inequality is due to the positiveness in equation 14. Since by definition

$$B_{\psi_t}(x^*, x_t) = \psi_t(x^*) - \psi_t(x_t) - \langle \nabla \psi_t(x_t), x^* - x_t \rangle$$
$$B_{\psi_t}(x^*, x_{t+1}) = \psi_t(x^*) - \psi_t(x_{t+1}) - \langle \nabla \psi_t(x_{t+1}), x^* - x_{t+1} \rangle \quad (16)$$
$$B_{\psi_t}(x_{t+1}, x_t) = \psi_t(x_{t+1}) - \psi_t(x_t) - \langle \nabla \psi_t(x_t), x_{t+1} - x_t \rangle$$

Therefore

$$
\begin{aligned}
&\alpha_t(f_t(x_t) - f_t(x^*)) + \alpha_t(\phi(x_{t+1}) - \phi(x^*)) \\
&\leq B_{\psi_t}(x^*, x_t) - B_{\psi_t}(x^*, x_{t+1}) - B_{\psi_t}(x_{t+1}, x_t) + \alpha_t \langle f_t'(x_t), x_t - x_{t+1} \rangle \\
&\leq B_{\psi_t}(x^*, x_t) - B_{\psi_t}(x^*, x_{t+1}) - B_{\psi_t}(x_{t+1}, x_t) + \frac{1}{2}\|x_t - x_{t+1}\|_{\psi_t}^2 + \frac{\alpha_t^2}{2}\|f_t'(x_t)\|_{\psi_t^*}^2 \quad (17) \\
&\leq B_{\psi_t}(x^*, x_t) - B_{\psi_t}(x^*, x_{t+1}) + \frac{\alpha_t^2}{2}\|f_t'(x_t)\|_{\psi_t^*}^2
\end{aligned}
$$

where the second inequality is due to the Fenchel's inequality on the conjugate functions $\frac{1}{2}\|\cdot\|_{\psi_t}^2$ and $\frac{1}{2}\|\cdot\|_{\psi_t^*}^2$. The last inequality is due to the strong convexity of $B_{\psi_t}(\cdot, \cdot)$. Therefore the lemma can be proved by dividing $\alpha_T$ on both sides and take the summation.

### A.3  PROOF OF THEOREM 2.1

**Proof.** When $\psi_t(x) = \langle x, H_t x \rangle$, the dual norm $\|\cdot\|_{\psi_t^*}$ is the Mahalanobis norm $\|\cdot\|_{H_t^{-1}}$

$$
\begin{aligned}
B_{\psi_t}(x, y) &= \psi_t(x) - \psi_t(y) - \langle \nabla \psi_t(y), x - y \rangle \\
&= \langle x, H_t x \rangle - \langle y, H_t y \rangle - 2\langle H_t y, x - y \rangle \quad (18) \\
&= \langle (x - y), H_t(x - y) \rangle
\end{aligned}
$$

Therefore by Lemma A.1,

$$
\begin{aligned}
&\sum_{t=1}^{T} f_t(x_t) - f_t(x^*) + \phi(x_t) - \phi(x^*) \\
&\leq \sum_{t=1}^{T} \frac{B_{\psi_t}(x^*, x_t)}{\alpha_t} - \frac{B_{\psi_t}(x^*, x_{t+1})}{\alpha_t} + \frac{\alpha_t}{2}\|f_t'(x_t)\|_{\psi_t^*}^2 \\
&= \sum_{t=1}^{T} \frac{\|H_t^{1/2}(x_t - x^*)\|^2}{\alpha_t} - \frac{\|H_t^{1/2}(x_{t+1} - x^*)\|^2}{\alpha_t} + \frac{\alpha_t}{2}\|f_t'(x_t)\|_{\psi_t^*}^2 \\
&\leq \frac{\|H_1^{1/2}(x_1 - x^*)\|^2}{\alpha} + \sum_{t=2}^{T} \frac{\|H_t^{1/2}(x_t - x^*)\|^2}{\alpha_t} - \frac{\|H_{t-1}^{1/2}(x_t - x^*)\|^2}{\alpha_{t-1}} + \sum_{t=1}^{T} \frac{\alpha_t}{2}\|f_t'(x_t)\|_{\psi_t^*}^2 \\
&\leq \sum_{i=1}^{d} \frac{h_{1,i}}{\alpha_1}(x_{1,i} - x_i^*)^2 + \sum_{t=2}^{T}\sum_{i=1}^{d}(\frac{h_{t,i}}{\alpha_t} - \frac{h_{t-1,i}}{\alpha_{t-1}})(x_{t,i} - x_i^*)^2 + \sum_{t=1}^{T} \frac{\alpha_t}{2}\|f_t'(x_t)\|_{\psi_t^*}^2 \\
&= \sum_{t=1}^{T}\sum_{i=1}^{d}(\frac{h_{t,i}}{\alpha_t} - \frac{h_{t-1,i}}{\alpha_{t-1}})(x_{t,i} - x_i^*)^2 + \sum_{t=1}^{T}\sum_{i=1}^{d} \frac{\alpha_t}{2}\frac{\alpha_t f_{t,i}'(x_t)^2}{2h_{t,i}} \\
&\leq \sum_{t=1}^{T}\sum_{i=1}^{d}(\frac{h_{t,i}}{\alpha_t} - \frac{h_{t-1,i}}{\alpha_{t-1}})D_{t,\infty}^2 + \sum_{t=1}^{T}\sum_{i=1}^{d} \frac{\alpha_t f_{t,i}'(x_t)^2}{2h_{t,i}}
\end{aligned}
$$

$$(19)$$

where the second inequality is by re-arranging the sum and deleting the last negative term.

## A.4 Proof of Theorem 4.1

**Proof.** Based on Theorem 2.1 and Proposition 4.1, take $h_t$ to be the design in Algorithm 1 with $c_t = 1$. Then after the final $\tau_{m_i}^{(i)}$ (which means the last gradient that is "large") for each dimension $i$, the first term doesn't increase and the second term can be bounded as follows.

$$
\begin{aligned}
R(T) &\leq \sum_{t=1}^{T}\sum_{i=1}^{d}\left(\frac{h_{t,i}}{\alpha_t} - \frac{h_{t-1,i}}{\alpha_{t-1}}\right)D_{t,\infty}^2 + \sum_{t=1}^{T}\sum_{i=1}^{d}\frac{\alpha_t g_{t,i}^2}{2h_{t,i}} \\
&\leq \sum_{i=1}^{d}\frac{h_{\tau_{m_i}^{(i)},i}}{\alpha_{\tau_{m_i}^{(i)}}}D_\infty^2 + \sum_{t=1}^{T}\sum_{i=1}^{d}\frac{\alpha_t}{2}|g_{t,i}| \\
&= \sum_{i=1}^{d}\frac{h_{\tau_{m_i}^{(i)},i}}{\alpha_{\tau_{m_i}^{(i)}}}D_\infty^2 + \sum_{t=1}^{\tau_{m_i}^{(i)}}\sum_{i=1}^{d}\frac{\alpha_t}{2}|g_{t,i}| + \sum_{i=1}^{d}\sum_{t=\tau_{m_i}^{(i)}+1}^{T}\frac{\alpha_t}{2}|g_{t,i}| \\
&\leq \sum_{i=1}^{d}\frac{h_{\tau_{m_i}^{(i)},i}\sqrt{\tau_{m_i}^{(i)}}}{\alpha}D_\infty^2 + \sum_{t=1}^{\tau_{m_i}^{(i)}}\sum_{i=1}^{d}\frac{\alpha_t}{2}|g_{t,i}| + \sum_{i=1}^{d}\sum_{t=\tau_{m_i}^{(i)}+1}^{T}\frac{\alpha}{2}\frac{\sqrt{\tau_{m_i}^{(i)}}}{t}|g_{\tau_{m_i}^{(i)},i}| \\
&\leq \sum_{i=1}^{d}\frac{h_{\tau_{m_i}^{(i)},i}\sqrt{\tau_{m_i}^{(i)}}}{\alpha}D_\infty^2 + \sum_{t=1}^{\tau_{m_i}^{(i)}}\sum_{i=1}^{d}\frac{\alpha_t}{2}|g_{t,i}| + \sum_{i=1}^{d}\frac{\alpha}{2}\sqrt{\tau_{m_i}^{(i)}}|g_{\tau_{m_i}^{(i)},i}|\log\left(\frac{T}{\tau_{m_i}^{(i)}}\right) \\
&\leq \sum_{i=1}^{d}\frac{D_\infty^2\sqrt{\tau_{m_i}^{(i)}}}{\alpha}h_{\tau_{m_i}^{(i)},i} + \sum_{i=1}^{d}\frac{\alpha}{2}\sqrt{1+\log\tau_{m_i}^{(i)}}\|g_{1:\tau_{m_i}^{(i)},i}\|_2 + \sum_{i=1}^{d}\frac{\alpha}{2}\sqrt{\tau_{m_i}^{(i)}}|g_{\tau_{m_i}^{(i)},i}|\log\left(\frac{T}{\tau_{m_i}^{(i)}}\right)
\end{aligned}
\tag{20}
$$

where the first inequality is by Theorem 2.1 and the second one is by Proposition 4.1. The third inequality is by the fact that $|g_{t,i}| \leq \sqrt{\frac{\tau_{m_i}^{(i)}}{t}}h_{\tau_{m_i}^{(i)},i} = \sqrt{\frac{\tau_{m_i}^{(i)}}{t}}|g_{\tau_{m_i}^{(i)},i}|$ for all $t \geq \tau_{m_i}^{(i)} + 1$. The fourth inequality is by $\sum_{t=\tau_{m_i}^{(i)}+1}^{T}1/t \leq \int_{t=\tau_{m_i}^{(i)}}^{T}1/t = \log\left(\frac{T}{\tau_{m_i}^{(i)}}\right)$. The last inequality is by the fact that

$$
\begin{aligned}
\sum_{t=1}^{\tau_{m_i}^{(i)}}\sum_{i=1}^{d}\frac{\alpha_t}{2}|g_{t,i}| &\leq \sum_{i=1}^{d}\|g_{1:\tau_{m_i}^{(i)},i}\|_2\sqrt{\sum_{t=1}^{\tau_{m_i}^{(i)}}\frac{1}{t}} \\
&\leq \sum_{i=1}^{d}\frac{\alpha}{2}\sqrt{1+\log\tau_{m_i}^{(i)}}\|g_{1:\tau_{m_i}^{(i)},i}\|_2
\end{aligned}
\tag{21}
$$

where the first inequality is by the Cauchy-Schwarz Inequality.

## A.5 Discussions about the Possible Designs of $c_t$

We first show our claim that given a design of $c_t$, scaling it by a constant $a$ is the same as scaling the step size $\alpha_t$ by $1/a$. Note that the maximum operation can be commuted with constant scaling, therefore by unrolling the maximum operation

$$
\begin{aligned}
h_t &= \max\left\{\sqrt{\frac{t-1}{t}}h_{t-1}, ac_t|g_t|\right\} \\
&= a \cdot \max\left\{\frac{1}{a}\sqrt{\frac{t-1}{t}}h_{t-1}, c_t|g_t|\right\} \\
&= a \cdot \max\left\{\frac{1}{a}\sqrt{\frac{t-1}{t}}\max\left\{\sqrt{\frac{t-2}{t-1}}h_{t-2}, ac_{t-1}|g_{t-1}|\right\}, c_t|g_t|\right\} \\
&= a \cdot \max\left\{\frac{1}{a}\sqrt{\frac{t-2}{t}}h_{t-2}, \sqrt{\frac{t-1}{t}}c_{t-1}|g_{t-1}|, c_t|g_t|\right\} \\
&= \cdots \\
&= a \cdot \max\left\{\sqrt{\frac{t-j}{t}}c_{t-j}|g_{t-j}|\right\}_{j=0}^{t-1}
\end{aligned}
\tag{22}
$$

Therefore we know that it is equivalent to scaling $\alpha_t$ by $1/a$ since it magnifies all $c_t|g_t|$ at the same time. Now given the discussions in section 4, we know that $c_t$ can be at most a constant, which is because it is strictly $\Omega(1)$, then the regret is strictly $\Omega(\sqrt{T})$. Now we can possibly use a decaying $c_t$. However, if $c_t$ is too small, then the regret may also be strictly $\Omega(\sqrt{T})$ because the second term in Theorem 2.1 can be very large. We show that $c_t = \Omega(1/\sqrt{t})$ by giving a counter example of the sequence of $g_t$ such that the regret becomes very large. Suppose $|g_t| = (1/t)^{1/4}|g_1|, |g_{1,i}| > 0, \forall i, c_t = 1/\sqrt{t}$, then since the size of the gradients keep decreasing, hence

$$
\begin{aligned}
h_t &= \max\left\{\sqrt{\frac{t-j}{t}}c_{t-j}|g_{t-j}|\right\}_{j=0}^{t-1} \\
&= \max\left\{\sqrt{\frac{t-j}{t}}\frac{1}{\sqrt{t-j}}|g_{t-j}|\right\}_{j=0}^{t-1} \\
&= \frac{1}{\sqrt{t}}\max\left\{|g_{t-j}|\right\}_{j=0}^{t-1} \\
&= \frac{1}{\sqrt{t}}|g_1|
\end{aligned}
\tag{23}
$$

The specific distribution of gradients meets the requirement for adaptive algorithms to work well because $\|g_{1:T,i}\|_2 = |g_{1,i}|\sqrt{\sum_{t=1}^{T}\frac{1}{\sqrt{t}}} = O(T^{1/4})$. Now, despite the first term in the regret bound in Theorem 2.1 becomes a constant, the second term becomes very large because

$$
\begin{aligned}
R(T) &\leq \sum_{t=1}^{T}\sum_{i=1}^{d}\left(\frac{h_{t,i}}{\alpha_t} - \frac{h_{t-1,i}}{\alpha_{t-1}}\right)D_{t,\infty}^2 + \sum_{t=1}^{T}\sum_{i=1}^{d}\frac{\alpha_t g_{t,i}^2}{2h_{t,i}} \\
&= \sum_{i=1}^{d}\frac{h_{1,i}}{\alpha_1}D_{\infty}^2 + \sum_{t=1}^{T}\sum_{i=1}^{d}\frac{\alpha g_{1,i}^2}{2\sqrt{t}|g_{1,i}|} \\
&= \sum_{i=1}^{d}\frac{h_{1,i}}{\alpha_1}D_{\infty}^2 + \sum_{i=1}^{d}\sum_{t=1}^{T}\frac{\alpha|g_{1,i}|}{2\sqrt{t}}
\end{aligned}
\tag{24}
$$

Now notice that

$$\sqrt{T} \le \sum_{t=1}^{T} \frac{1}{\sqrt{t}} \le 2\sqrt{T} - 1 \tag{25}$$

Therefore the regret upper bound there is already $\Theta(\sqrt{T})$. Changing the gradients to be further larger, for example, $|g_t| = (1/t)^{1/8}|g_1|$, will still satisfy the $\|g_{1:T,i}\|_2 \ll \sqrt{T}$ condition, but also make the regret bound even larger ($\Theta(T^{3/4})$). Therefore using $c_t = O(1/\sqrt{t})$ is unacceptable in terms of the regret bound. Note that we can of course change the initial time step, the argument still holds if the first few $c_t$'s are not smaller than $\Theta(1/\sqrt{t})$ in order. Next, we show the regret bound is indeed achievable by the regret. We propose the following theorem:

**Theorem A.1** *If $c_t = O(1/\sqrt{t})$ in Algorithm 1, then there exists an online convex optimization problem where $\|g_{1:T,i}\|_2 \ll \sqrt{T}$ and AMX has a regret of size $\Omega(\sqrt{T})$*

**Proof.** We recall the terms in the proof of Lemma A.1.

$$\begin{aligned}
&\alpha_t(f_t(x_t) - f_t(x^*)) + \alpha_t(\phi(x_{t+1}) - \phi(x^*)) \\
&\le \alpha_t \langle f_t'(x_t), x_t - x^* \rangle + \alpha_t \langle \phi_t'(x_{t+1}), x_{t+1} - x^* \rangle \\
&= \alpha_t \langle f_t'(x_t), x_{t+1} - x^* \rangle + \alpha_t \langle \phi_t'(x_{t+1}), x_{t+1} - x^* \rangle + \alpha_t \langle f_t'(x_t), x_t - x_{t+1} \rangle
\end{aligned} \tag{26}$$

If we set $\phi(x) = 0$ in the above inequality and let $f_t(x) = g_t x$ be a linear loss function, then the first inequality becomes an equality. Moreover, it's possible to make the term $\alpha_t \langle f_t'(x_t), x_{t+1} - x^* \rangle > 0$ for all $t$, as long as we change the loss function to be positive when $x_{t+1} > x^*$ and negative when $x_{t+1} < x^*$, then we know the regret is

$$\begin{aligned}
R(T) &= \sum_{t=1}^{T} f_t(x_t) - f_t(x^*) \\
&= \sum_{t=1}^{T} \langle f_t'(x_t), x_{t+1} - x^* \rangle + \langle f_t'(x_t), x_t - x_{t+1} \rangle \\
&\ge \sum_{t=1}^{T} \langle f_t'(x_t), x_t - x_{t+1} \rangle \\
&= \sum_{t=1}^{T} \langle g_t, \frac{\alpha_t}{2} H_t^{-1} g_t \rangle \\
&= \sum_{t=1}^{T} \sum_{i=1}^{d} \frac{\alpha_t g_{t,i}^2}{2 h_{t,i}}
\end{aligned} \tag{27}$$

Notice what we have claimed before Theorem A.1 are all about this term being larger than $O(\sqrt{T})$, therefore we prove our claim.

## B  FULL MATRIX PROXIMAL FUNCTION

When $H_t$ is not a diagonal matrix but a full matrix, we can similarly prove a general regret bound for any matrix in Theorem B.1 below.

**Theorem B.1** *Let the sequence $\{x_t\}$ be defined by the update rule (1) and for any $x^*$, denote $D_{t,2}^2 = \|x_t - x^*\|_2^2$. When $\psi_t(x) = \langle x, H_t x \rangle$, where $H_t \in S_d^+$ is a general matrix, assume without loss of generality that $\phi(x_1) = 0, H_0 = 0$, if $tr(\frac{H_t}{\alpha_t}) \ge tr(\frac{H_{t-1}}{\alpha_{t-1}})$, then*

$$R(T) \le \sum_{t=1}^{T} D_{t,2}^2 tr(\frac{H_t}{\alpha_t} - \frac{H_{t-1}}{\alpha_{t-1}}) + \sum_{t=1}^{T} \frac{\alpha_t}{2} g_t^T H_t^{-1} g_t. \tag{28}$$

**Proof**. When $H_t$ is a full matrix, similar to Theorem 2.1, we can get the regret bound

$$R(T) \le \sum_{t=1}^{T} \frac{B_{\psi_t}(x^*, x_t)}{\alpha_t} - \frac{B_{\psi_{t-1}}(x^*, x_t)}{\alpha_{t-1}} + \frac{\alpha_t}{2} \langle g_t, H_t^{-1} g_t \rangle \tag{29}$$

Let $\lambda_{\max}(M)$ denote the largest eigenvalue of a matrix $M$, then

$$\frac{B_{\psi_t}(x^*, x_t)}{\alpha_t} - \frac{B_{\psi_{t-1}}(x^*, x_t)}{\alpha_{t-1}} = \langle x^* - x_t, (\frac{H_t}{\alpha_t} - \frac{H_{t-1}}{\alpha_{t-1}})(x^* - x_t) \rangle$$
$$\le \|x^* - x_t\|_2^2 \lambda_{\max}(\frac{H_t}{\alpha_t} - \frac{H_{t-1}}{\alpha_{t-1}}) \le \|x^* - x_t\|_2^2 \text{tr}(\frac{H_t}{\alpha_t} - \frac{H_{t-1}}{\alpha_{t-1}}) \tag{30}$$

Hence we prove the regret bound shown in Theorem B.1. Moreover, the marginal regret bound minimization problem in this case corresponding to Theorem B.1 and the constraint is

$$\min_{H_t} D_{t,2}^2 \text{tr}(\frac{H_t}{\alpha_t} - \frac{H_{t-1}}{\alpha_{t-1}}) + \frac{\alpha_t}{2} \text{tr}(H_t^{-1} g_t g_t^T), \text{ s.t.} H_t \succ 0, \text{tr}(\frac{H_t}{\alpha_t}) \ge \text{tr}(\frac{H_{t-1}}{\alpha_{t-1}}) \tag{31}$$

where $D_{t,2} = \|x_t - x^*\|_2$. Now, we propose a proposition that shows the optimal full matrix solution is the one that reaches the infimum of the problem.

**Proposition B.1** *The following matrix*

$$H_t^* = max \left\{ \frac{tr(\sqrt{\frac{t-1}{t}} H_{t-1})}{tr((\frac{\alpha_t^2}{2D_{t,2}^2} g_t g_t^T)^{1/2})}, 1 \right\} (\frac{\alpha_t^2}{2D_{t,2}^2} g_t g_t^T)^{1/2}, \quad \textit{(Full)} \tag{32}$$

*and its Moore-Penrose pseudoinverse $H_t^{*-}$ gives an infimum to the problem (31). i.e.*

$$D_{t,2}^2 tr(\frac{H_t^*}{\alpha_t}) + \frac{\alpha_t}{2} tr(H_t^{*-} g_t g_t^T) = inf_{H_t \succ 0, tr(\frac{H_t}{\alpha_t}) \ge tr(\frac{H_{t-1}}{\alpha_{t-1}})} \left\{ D_{t,2}^2 tr(\frac{H_t}{\alpha_t}) + \frac{\alpha_t}{2} tr(H_t^{-1} g_t g_t^T) \right\} \tag{33}$$

**Proof.** Now similarly, we can first construct the Lagrangian for the marginal regret bound minimization problem. Let $\theta \ge 0$ denote a Lagrangian parameter for the trace constraint, and $Z \succeq 0$ for the positive definiteness constraint. Then the Lagrangian problem is

$$L(H_t, \theta, Z) = D_{t,2}^2 \text{tr}(\frac{H_t}{\alpha_t}) + \frac{\alpha_t}{2} \text{tr}(H_t^{-1} g_t g_t^T) - \theta \text{tr}((\frac{H_t}{\alpha_t} - \frac{H_{t-1}}{\alpha_{t-1}})) - \text{tr}(H_t Z) \tag{34}$$

Take derivative with respect to $H_t$ we can get

$$\frac{D_{t,2}^2}{\alpha_t} I - \frac{\alpha_t}{2} H_t^{-1} g_t g_t^T H_t^{-1} - \theta I - Z = 0 \tag{35}$$

where $I$ is the identity matrix. If an invertible $H_t$ can be found, then the generalized complementarity condition (Boyd & Vandenberghe, 2004) implies that $Z = 0$ and either $\theta = 0$ or $\text{tr}(\frac{H_t}{\alpha_t}) = \text{tr}(\frac{H_{t-1}}{\alpha_{t-1}})$.

If $\theta = 0$

$$H_t = (\frac{\alpha_t^2}{2D_{t,2}^2} g_t g_t^T)^{1/2} \text{ and tr}(\frac{H_t}{\alpha_t}) \ge \text{tr}(\frac{H_{t-1}}{\alpha_{t-1}}) \tag{36}$$

However, this is not an acceptable solution since it is not invertible.

If $\theta \ne 0$, note that we need $\theta$ to be real, hence there is no solution because $g_t g_t^T$ has rank at most 1, and even if $\theta = D_{t,2}^2/\alpha_t$, there does not exist a matrix that makes the term $H_t^{-1} g_t g_t^T H_t^{-1}$ zero.

Instead, we propose the following matrix reaches the infimum of problem 31.

$$H_t^* = \max \left\{ \frac{\text{tr}(\sqrt{\frac{t-1}{t}} H_{t-1})}{\text{tr}((\frac{\alpha_t^2}{2D_{t,2}^2} g_t g_t^T)^{1/2})}, 1 \right\} (\frac{\alpha_t^2}{2D_{t,2}^2} g_t g_t^T)^{1/2} \tag{37}$$

It can be understood as a special maximum operation that makes sure

$$\text{tr}(H_t^*) = \max \left\{ \text{tr}(\sqrt{\frac{t-1}{t}} H_{t-1}), \frac{\alpha_t}{\sqrt{2} D_{t,2}} \text{tr}((g_t g_t^T)^{1/2}) \right\}$$

Now, a very important remark here is that the solution is not invertible, because $g_t g_t^T$ has rank at most . Therefore, the equation above is not acceptable as a direct solution to problem (31). However, setting $H_t$ to be as in the above equation gives a solution to the infimum problem in **Proposition B.1**. Let $g_t g_t^T$ be diagonally decomposed by

$$g_t g_t^T = Q \begin{bmatrix} v & 0 \\ 0 & 0 \end{bmatrix} Q^T, v = g_t^T g_t = \sum g_{t,i}^2 \tag{38}$$

and define the matrices $H_t(\delta)$ that can be written as,

$$H_t(\delta) = \max \left\{ \frac{\text{tr}(\sqrt{\frac{t-1}{t}} H_{t-1})}{\sqrt{v} + n\delta}, \frac{\alpha_t \sqrt{v}}{\sqrt{2} D_{t,2}(\sqrt{v} + n\delta)} \right\} Q \begin{bmatrix} \sqrt{v} & 0 \\ 0 & \delta I \end{bmatrix} Q^T, \tag{39}$$

Therefore we know that $\lim_{\delta \to 0} H_t(\delta) = H_t^*$ and that

$$D_{t,2}^2 \text{tr}(\frac{H_t(\delta)}{\alpha_t}) + \frac{\alpha_t}{2} \text{tr}(H_t(\delta)^{-1} g_t g_t^T)$$

$$= D_{t,2}^2 \max \left\{ \text{tr}(\frac{\sqrt{t-1}}{\alpha} H_{t-1}), \frac{\sqrt{v}}{\sqrt{2} D_{t,2}} \right\} + \min \left\{ \frac{\alpha_t(\sqrt{v} + n\delta)}{2\text{tr}(\sqrt{\frac{t-1}{t}} H_{t-1})}, \frac{\sqrt{2} D_{t,2}(\sqrt{v} + n\delta)}{2\sqrt{v}} \right\} \text{tr}(Q \begin{bmatrix} \sqrt{v} & 0 \\ 0 & 0 \end{bmatrix} Q^T)$$

$$= D_{t,2}^2 \max \left\{ \text{tr}(\frac{\sqrt{t-1}}{\alpha} H_{t-1}), \frac{\sqrt{v}}{\sqrt{2} D_{t,2}} \right\} + \min \left\{ \frac{(\sqrt{v} + n\delta)\sqrt{v}\alpha_t}{2\text{tr}(\sqrt{\frac{t-1}{t}} H_{t-1})}, \frac{\sqrt{2} D_{t,2}(\sqrt{v} + n\delta)}{2} \right\}$$

$$\tag{40}$$

Hence

$$\lim_{\delta \to 0} D_{t,2}^2 \text{tr}(\frac{H_t(\delta)}{\alpha_t}) + \frac{\alpha_t}{2} \text{tr}(H_t(\delta)^{-1} g_t g_t^T)$$

$$= D_{t,2}^2 \max \left\{ \text{tr}(\frac{\sqrt{t-1}}{\alpha} H_{t-1}), \frac{\sqrt{v}}{\sqrt{2} D_{t,2}} \right\} + \min \left\{ \frac{\alpha_t v}{2\text{tr}(\sqrt{\frac{t-1}{t}} H_{t-1})}, \frac{\sqrt{2} D_{t,2}(\sqrt{v})}{2} \right\}$$

$$= D_{t,2}^2 \text{tr}(\frac{H_t^*}{\alpha_t}) + \frac{\alpha_t}{2} \text{tr}(H_t^{*-} g_t g_t^T) \tag{41}$$

$$= \begin{cases} D_{t,2}^2 \text{tr}(\frac{\sqrt{t-1}}{\alpha} H_{t-1}) + \frac{\alpha_t v}{2\text{tr}(\sqrt{\frac{t-1}{t}} H_{t-1})} & \text{if } \text{tr}(\sqrt{\frac{t-1}{t}} H_{t-1}) \text{ is larger} \\ \\ \sqrt{2} D_{t,2} \sqrt{v} & \text{if } \frac{\alpha_t}{\sqrt{2} D_{t,2}} \text{tr}((g_t g_t^T)^{1/2}) \text{ is larger} \end{cases}$$

Now, let $g(\theta) = \inf_{H_t}(L(H_t, \theta, Z(\theta)))$ be the dual of problem (31), where when $\frac{D_{t,2}^2}{\alpha_t} > \theta$ we define the matrices $Z(\theta)$ and $H_t(\theta, \delta)$ as

$$Z(\theta) = \begin{bmatrix} 0 & 0 \\ 0 & (\frac{D_{t,2}^2}{\alpha_t} - \theta)I \end{bmatrix}, H_t(\theta, \delta) = \frac{\alpha_t}{\sqrt{2(D_{t,2}^2 - \theta\alpha_t)}} Q \begin{bmatrix} \sqrt{v} & 0 \\ 0 & \delta I \end{bmatrix} Q^T, \qquad (42)$$

Then from the derivative with respect to $H_t$, we know that

$$(\frac{D_{t,2}^2}{\alpha_t} - \theta)I - \frac{\alpha_t}{2} H_t(\theta, \delta)^{-1} g_t g_t^T H_t(\theta, \delta)^{-1} - Z(\theta) = 0 \qquad (43)$$

Therefore $H_t(\theta, \delta)$ achieves the minimum in the dual, moreover

$$g(\theta) = D_{t,2}^2 \text{tr}(\frac{H_t(\theta, \delta)}{\alpha_t}) + \frac{\alpha_t}{2}\text{tr}(H_t(\theta, \delta)^{-1} g_t g_t^T) - \theta\text{tr}((\frac{H_t(\theta, \delta)}{\alpha_t} - \frac{H_{t-1}}{\alpha_{t-1}})) - \text{tr}(H_t(\theta, \delta)Z(\theta))$$

$$= D_{t,2}^2 \frac{(\sqrt{v} + \delta n)}{\sqrt{2(D_{t,2}^2 - \theta\alpha_t)}} + \frac{\sqrt{2v(D_{t,2}^2 - \theta\alpha_t)}}{2} - \theta(\frac{(\sqrt{v} + \delta n)}{\sqrt{2(D_{t,2}^2 - \theta\alpha_t)}} - \text{tr}(\frac{H_{t-1}}{\alpha_{t-1}})) - \frac{(n-1)\delta(D_{t,2}^2 - \theta\alpha_t)}{\sqrt{2(D_{t,2}^2 - \theta\alpha_t)}}$$

$$= (D_{t,2}^2 - \theta)\frac{(\sqrt{v} + \delta n)}{\sqrt{2(D_{t,2}^2 - \theta\alpha_t)}} + \frac{\sqrt{2v(D_{t,2}^2 - \theta\alpha_t)}}{2} + \theta\text{tr}(\frac{H_{t-1}}{\alpha_{t-1}}) - \frac{(n-1)\delta\sqrt{D_{t,2}^2 - \theta\alpha_t}}{\sqrt{2}}$$

$$\qquad (44)$$

Notice that take $\theta = 0$, $\lim_{\delta \to 0} g(\theta) = \sqrt{2}D_{t,2}\sqrt{v}$, and take $\theta$ to be

$$\sqrt{2v(D_{t,2}^2 - \theta\alpha_t)} = \frac{\alpha_t(\sqrt{v} + n\delta)}{\text{tr}(\frac{\alpha_t}{\alpha_{t-1}} H_{t-1})} \qquad (45)$$

then

$$g(\theta) = D_{t,2}^2 \text{tr}(\frac{1}{\alpha_{t-1}} H_{t-1}) + \frac{\alpha_t v}{2\text{tr}(\sqrt{\frac{t-1}{t}} H_{t-1})} \qquad (46)$$

Note that they are equal to the two cases respectively, therefore the duality gap in this problem is zero and $H_t^*$ is indeed the infimum solution.

Similarly, $H_t$ is always better than any other full matrix proximal function at $t$ in terms of regret. A unified algorithm is provided in Algorithm 2, where the $\max^*$ operation is a special operation that shows how the diagonal AMX algorithm is similar to the full matrix one, in the sense that it tries to find the maximum trace between the two matrices, i.e.

$$\text{tr}(H_t^*) = \max\left\{\text{tr}(\sqrt{\frac{t-1}{t}} H_{t-1}), \text{tr}((c_t g_t g_t^T)^{1/2})\right\}$$

## C   DISCUSSIONS ON AMX ALGORITHM WITH MOMENTUM

We provide the detailed implementation of the diagonal AMX algorithm with momentum and decoupled weight decay here. The momentum term is implemented in line 5 in Algorithm 3, where $m_t = \beta_{1t}m_{t-1} + (1 - \beta_{1t})g_t$ and $\{\beta_{1t}\}_{t=1}^T$ are called the momentum parameters. This implementation is exactly the same as the implementation of modern adaptive algorithms such as Kingma & Ba (2015); Reddi et al. (2018); Huang et al. (2019) and Li et al. (2020).

---

**Algorithm 2** AMX Algorithm (Composite Mirror Descent Form)

1: **Input:** $x \in \mathcal{F}, \{\alpha_t\}_{t=1}^T, \{c_t\}_{t=1}^T, \phi(x)$
2: **Initialize** $h_0 = 0, H_0 = 0$
3: **for** $t = 1$ **to** $T$ **do**
4: $\quad g_t = \nabla f_t(x_t)$
5: $\quad$ **if** $H_t$ is a diagonal matrix **then**
6: $\quad\quad h_t = \max(\sqrt{\frac{t-1}{t}}h_{t-1}, (c_t g_t^2)^{1/2})$
7: $\quad\quad H_t = \text{diag}(h_t) + \epsilon$
8: $\quad$ **else**
9: $\quad\quad H_t = \max^*(\sqrt{\frac{t-1}{t}}H_{t-1}, (c_t g_t g_t^T)^{1/2}) + \epsilon I$
10: $\quad$ **end if**
11: $\quad x_{t+1} = \text{argmin}_{x \in \mathcal{X}}\{\alpha_t\langle g_t, x\rangle + \alpha_t\phi(x) + \langle x - x_t, H_t(x - x_t)\rangle\}$
12: **end for**

---

**Algorithm 3** AMX Algorithm with Momentum (Diagonal)

1: **Input:** $x \in \mathcal{F}, \{\alpha_t\}_{t=1}^T, \{\beta_{1t}\}_{t=1}^T, c_t = 1, \epsilon = 1e - 8, \lambda = 5e - 2$
2: **Initialize** $m_0 = 0, h_0 = 0$
3: **for** $t = 1$ **to** $T$ **do**
4: $\quad g_t = \nabla f_t(x_t)$
5: $\quad m_t = \beta_{1t}m_{t-1} + (1 - \beta_{1t})g_t$
6: $\quad h_t = \sqrt{\max(\frac{t-1}{t}h_{t-1}^2, c_t g_t^2) + \epsilon}$
7: $\quad H_t = \text{diag}(h_{t,1}, h_{t,2}, \cdots, h_{t,d})$
8: $\quad x_{t+1} = \Pi_{\mathcal{F}, H_t}(x_t - \alpha_t m_t/h_t - \lambda\alpha_t x_t)$
9: **end for**

---

The following theorem provides a regret bound for algorithm 3 and shows its convergence.

**Theorem C.1** *Let $\{x_t\}$ and $\{h_t\}$ be the sequences obtained from Algorithm 3, $\alpha_t = \alpha/\sqrt{t}, c_t = 1, \beta_{1,1} = \beta_1, \beta_{1,t} \leq \beta_1$, for all $t \in [T]$. Assume that $\mathcal{F}$ has bounded diameter $\|x - y\|_\infty \leq D_\infty, \forall x, y \in \mathcal{F}$ and $\|\nabla f_t(x)\| \leq G_\infty$ for all $t \in [T]$ and $x \in \mathcal{F}$. Then for $x_t$ generated using Algorithm 3, we have the following bound on the regret.*

$$R(T) \leq \frac{D_\infty^2\sqrt{T}}{2\alpha(1 - \beta_1)}\sum_{i=1}^d h_{T,i} + \frac{D_\infty^2}{2(1 - \beta_1)}\sum_{t=1}^T\sum_{i=1}^d\frac{\beta_{1t}h_{t,i}}{\alpha_t} + \frac{\alpha\sqrt{1 + \log T}}{(1 - \beta_1)^3}\sum_{i=1}^d\|g_{1:T,i}\|_2, \quad (47)$$

Corollary C.1 follows immediately from the above theorem.

**Corollary C.1** *Setting $\beta_{1t} = \beta_1\lambda^{t-1}$ in Theorem C.1, then we have*

$$R(T) \leq \frac{D_\infty^2\sqrt{T}}{2\alpha(1 - \beta_1)}\sum_{i=1}^d h_{T,i} + \frac{D_\infty^2 G_\infty}{2(1 - \beta_1)(1 - \lambda^2)} + \frac{\alpha\sqrt{1 + \log T}}{(1 - \beta_1)^3}\sum_{i=1}^d\|g_{1:T,i}\|_2 \quad (48)$$

Similarly, the bound above can be considered as better than the regret of SGD when $\sum_{i=1}^d h_{T,i} \ll \sqrt{d}$ and $\sum_{i=1}^d\|g_{1:T,i}\|_2 \ll \sqrt{dT}$ (Duchi et al., 2011), which is the same as the claims we make in Theorem 4.2. Another important observation is that Theorem 4.1 also holds here with similar arguments (because the first term is almost the same as the first term without momentum, the second term is a constant, and the third term is almost the same as well. We provide a similar theorem here.

**Theorem C.2** *Let $\{x_t\}$ and $\{h_t\}$ be the sequences obtained from Algorithm 3, $\alpha_t = \alpha/\sqrt{t}, c_t = 1, \beta_{1,1} = \beta_1, \beta_{1t} = \beta_1\lambda^{t-1}$, for all $t \in [T]$. Assume that $\mathcal{F}$ has bounded diameter $\|x - y\|_\infty \leq D_\infty, \forall x, y \in \mathcal{F}$ and $\|\nabla f_t(x)\| \leq G_\infty$ for all $t \in [T]$ and $x \in \mathcal{F}$. Let $\{\tau_{m_i}^{(i)}\}_{i=1}^d$ be the largest time steps $t$ on each dimension when $h_{t,i} = |g_{t,i}|$. Then for $x_t$ generated using Algorithm 3, we have the*

*following bound on the regret.*

$$R(T) \leq \frac{D_\infty^2}{2\alpha(1-\beta_1)} \sum_{i=1}^d h_{\tau_{m_i}^{(i)},i} \sqrt{\tau_{m_i}^{(i)}} + \frac{D_\infty^2 G_\infty}{2(1-\beta_1)(1-\lambda^2)} + \frac{\alpha}{(1-\beta_1)^3} \sum_{i=1}^d \sqrt{1 + \log \tau_{m_i}^{(i)}} \|g_{1:\tau_{m_i}^{(i)},i}\|_2,$$

$$+ \frac{\alpha}{(1-\beta_1)^3} \sum_{i=1}^d \sqrt{\tau_{m_i}^{(i)}} \log(\frac{T}{\tau_{m_i}^{(i)}}) |g_{\tau_{m_i}^{(i)},i}|$$

$$(49)$$

The above regret bound can be considered as significantly better than $O(\sqrt{T})$ because it mostly depends on the time steps $\tau_{m_i}^{(i)}$ instead of the number of time steps $T$. Given a distribution of gradients, it is possible that $\tau_{m_i}^{(i)}$'s are much smaller than $T$, and hence the regret is much smaller than common adaptive optimizers. The regret is at most $O(\sqrt{T})$, which is the same as AdaGrad and AMSGrad.

We also conducted experiments to examine the effect of momentum on the convergence and performance of our algorithm on CIFAR-10. As shown in figure 4, the choice of $\beta_1$ did not affect the convergence rate or the performance of AMX too much.

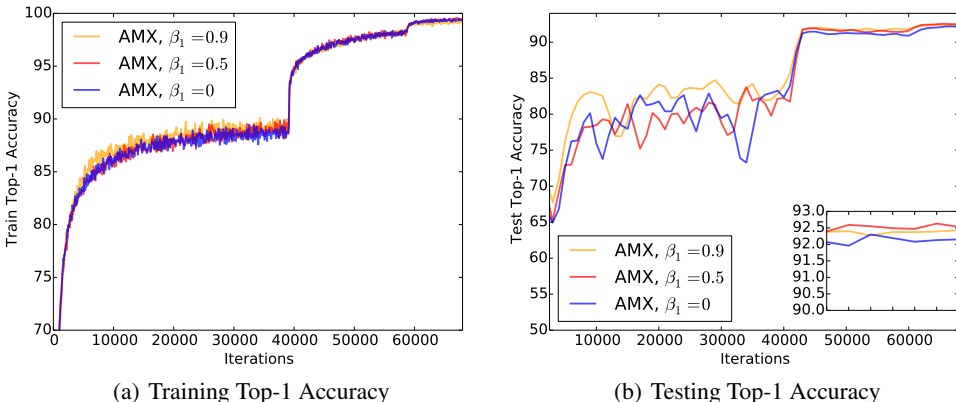

(a) Training Top-1 Accuracy        (b) Testing Top-1 Accuracy

Figure 4: Training and testing Top-1 accuracy curve on CIFAR-10 with different momentum parameters $\beta_1$

### C.1 PROOF OF REGRET BOUND OF AMX WITH MOMENTUM

#### C.1.1 PROOF OF THEOREM 4.2 AND THEOREM C.1

*Proof.* Following the proof by Reddi et al. (2018), we provide the proof of regret bound in Theorem 4.2 and Theorem C.1. Note that Algorithm 1 is a special case of algorithm 1. Therefore we only need to prove Theorem C.1 and Theorem 4.2 can be directly inferred by taking $\beta_{1t} = 0$ and change $\alpha$ by $\alpha/2$. By the definition of the projection operation $\Pi_{\mathcal{F},H_t}$, we know that

$$x_{t+1} = \Pi_{\mathcal{F},H_t}(x_t - \alpha_t H_t^{-1} m_t) = \operatorname{argmin}_{x \in \mathcal{F}} \|H_t^{1/2}(x - (x_t - \alpha_t H_t^{-1} m_t))\| \qquad (50)$$

Using Lemma 4 in Reddi et al. (2018) with a direct substitution of $z_1 = (x_t - \alpha_t H_t^{-1} m_t), Q = H_t$ and $z_2 = x^*$ for $x^* \in \mathcal{F}$, the following inequality holds:

$$\|H_t^{1/2}(u_1 - u_2)\|^2 = \|H_t^{1/2}(x_{t+1} - x^*)\|^2 \leq \|H_t^{1/2}(x_t - \alpha_t H_t^{-1} m_t - x^*)\|^2$$

$$= \|H_t^{1/2}(x_t - x^*)\|^2 + \alpha_t^2 \|H_t^{-1/2} m_t\|^2 - 2\alpha_t \langle m_t, (x_t - x^*) \rangle$$

$$= \|H_t^{1/2}(x_t - x^*)\|^2 + \alpha_t^2 \|H_t^{-1/2} m_t\|^2 - 2\alpha_t \langle \beta_{1t} m_{t-1} + (1-\beta_{1t})g_t, (x_t - x^*) \rangle$$

$$(51)$$

where the first equality is due to $\Pi_{\mathcal{F}, \sqrt{V_t}}(x^*) = x^*$. Rearrange the last inequality, we obtain

$$
\begin{aligned}
(1 - \beta_{1t})\langle g_t, (x_t - x^*)\rangle &\leq \frac{1}{2\alpha_t}\left[\|H_t^{1/2}(x_t - x^*)\|^2 - \|H_t^{1/2}(x_{t+1} - x^*)\|^2\right] + \frac{\alpha_t}{2}\|H_t^{-1/2}m_t\|^2 \\
&\quad - \beta_{1t}\langle m_{t-1}, (x_t - x^*)\rangle \\
&\leq \frac{1}{2\alpha_t}\left[\|H_t^{1/2}(x_t - x^*)\|^2 - \|H_t^{1/2}(x_{t+1} - x^*)\|^2\right] + \frac{\alpha_t}{2}\|H_t^{-1/2}m_t\|^2 \\
&\quad + \frac{\beta_{1t}\alpha_t}{2}\|H_t^{-1/2}m_{t-1}\|^2 + \frac{\beta_{1t}}{2\alpha_t}\|H_t^{1/2}(x_t - x^*)\|^2
\end{aligned}
$$
(52)

The second inequality is from applying the Cauchy-Schwarz and Young's inequality. By the convexity of $f_t$ and **Lemma C.1** and **Lemma C.2**, we have

$$
\begin{aligned}
\sum_{t=1}^{T} f_t(x_t) - f_t(x^*) &\leq \sum_{t=1}^{T}\langle g_t, (x_t - x^*)\rangle \\
&\leq \sum_{t=1}^{T}\frac{1}{2\alpha_t(1 - \beta_{1t})}\left[\|H_t^{1/2}(x_t - x^*)\|^2 - \|H_t^{1/2}(x_{t+1} - x^*)\|^2\right] + \frac{\alpha_t}{2(1 - \beta_{1t})}\|H_t^{-1/2}m_t\|^2 \\
&\quad + \frac{\beta_{1t}\alpha_t}{2(1 - \beta_{1t})}\|H_t^{-1/2}m_{t-1}\|^2 + \frac{\beta_{1t}}{2\alpha_t(1 - \beta_{1t})}\|H_t^{1/2}(x_t - x^*)\|^2 \\
&\leq \frac{D_{\infty}^2}{2\alpha_T(1 - \beta_1)}\sum_{i=1}^{d}h_{T,i} + \sum_{t=1}^{T}\frac{\beta_{1t}}{2\alpha_t(1 - \beta_1)}\|H_t^{1/2}(x_t - x^*)\|^2 + \frac{\alpha\sqrt{1 + \log T}}{(1 - \beta_1)^3}\sum_{i=1}^{d}\|g_{1:T,i}\|_2 \\
&= \frac{D_{\infty}^2}{2\alpha_T(1 - \beta_1)}\sum_{i=1}^{d}h_{T,i} + \sum_{t=1}^{T}\frac{1}{2\alpha_t(1 - \beta_1)}\sum_{i=1}^{d}\beta_{1t}(x_{t,i} - x_i^*)^2 h_{t,i} + \frac{\alpha\sqrt{1 + \log T}}{(1 - \beta_1)^3}\sum_{i=1}^{d}\|g_{1:T,i}\|_2 \\
&\leq \frac{D_{\infty}^2}{2\alpha_T(1 - \beta_1)}\sum_{i=1}^{d}h_{T,i} + \frac{D_{\infty}^2}{2(1 - \beta_1)}\sum_{t=1}^{T}\sum_{i=1}^{d}\frac{\beta_{1t}h_{t,i}}{\alpha_t} + \frac{\alpha\sqrt{1 + \log T}}{(1 - \beta_1)^3}\sum_{i=1}^{d}\|g_{1:T,i}\|_2 \\
&\leq \frac{D_{\infty}^2}{2\alpha_T(1 - \beta_1)}\sum_{i=1}^{d}h_{T,i} + \frac{D_{\infty}^2}{2(1 - \beta_1)}\sum_{t=1}^{T}\sum_{i=1}^{d}\frac{\beta_{1t}h_{t,i}}{\alpha_t} + \frac{\alpha\sqrt{1 + \log T}}{(1 - \beta_1)^3}\sum_{i=1}^{d}\|g_{1:T,i}\|_2
\end{aligned}
$$
(53)

To obtain the regret bound in 4.2, we can take $\beta_1 = 0$ and $\alpha = \alpha/2$ by equivalence in A.1 in the above bound, and the regret bound follows

$$
R_T \leq \frac{D_{\infty}^2\sqrt{T}}{\alpha}\sum_{i=1}^{d}h_{T,i} + \frac{\alpha}{2}\sqrt{1 + \log T}\sum_{i=1}^{d}\|g_{1:T,i}\|_2,
$$
(54)

### C.1.2 AUXILLARY LEMMAS

**Lemma C.1** *For the parameter settings and conditions assumed in **Theorem C.1**, we have*

$$
\sum_{t=1}^{T}\frac{\beta_{1t}\alpha_t}{2(1 - \beta_{1t})}\|H_t^{-1/2}m_t\|^2 \leq \frac{\alpha\sqrt{1 + \log T}}{(1 - \beta_1)^2}\sum_{i=1}^{d}\|g_{1:T,i}\|_2
$$
(55)

*where $C$ is a constant.*

*Proof:* We first analyze with the following process directly from the update rules, note that

$$
\begin{aligned}
m_{T,i} &= \sum_{j=1}^{T}(1 - \beta_{1j})\Pi_{k=1}^{T-j}\beta_{1(T-k+1)}g_{j,i} \\
h_{T,i}^2 &= \max(\frac{T-1}{T}h_{T-1,i}^2, g_{T,i}^2)
\end{aligned}
$$
(56)

$$\sum_{t=1}^{T} \alpha_t \|H_t^{-1/2} m_t\|^2$$

$$= \sum_{t=1}^{T-1} \alpha_t \|H_t^{-1/2} m_t\|^2 + \alpha_T \sum_{i=1}^{d} \frac{m_{T,i}^2}{h_{T,i}}$$

$$= \sum_{t=1}^{T-1} \alpha_t \|H_t^{-1/2} m_t\|^2 + \alpha \sum_{i=1}^{d} \frac{(\sum_{j=1}^{T} (1 - \beta_{1j}) \Pi_{k=1}^{T-j} \beta_{1(T-k+1)} g_{j,i})^2}{\sqrt{T \max(\frac{T-1}{T} h_{T-1,i}^2, g_{T,i}^2)}}$$

$$\leq \sum_{t=1}^{T-1} \alpha_t \|H_t^{-1/2} m_t\|^2 + \alpha \sum_{i=1}^{d} \frac{(\sum_{j=1}^{T} \Pi_{k=1}^{T-j} \beta_{1(T-k+1)})(\sum_{j=1}^{T} (1 - \beta_{1j})^2 \Pi_{k=1}^{T-j} \beta_{1(T-k+1)} g_{j,i}^2)}{\sqrt{T \max(\frac{T-1}{T} h_{T-1,i}^2, g_{T,i}^2)}}$$

$$\leq \sum_{t=1}^{T-1} \alpha_t \|H_t^{-1/2} m_t\|^2 + \alpha \sum_{i=1}^{d} \frac{(\sum_{j=1}^{T} \beta_1^{T-j})(\sum_{j=1}^{T} \beta_1^{T-j} g_{j,i}^2)}{\sqrt{T \max(\frac{T-1}{T} h_{T-1,i}^2, g_{T,i}^2)}}$$

$$\leq \sum_{t=1}^{T-1} \alpha_t \|H_t^{-1/2} m_t\|^2 + \frac{\alpha}{(1 - \beta_1)} \sum_{i=1}^{d} \frac{\sum_{j=1}^{T} \beta_1^{T-j} g_{j,i}^2}{\sqrt{T \max(\frac{T-1}{T} h_{T-1,i}^2, g_{T,i}^2)}}$$

$$\leq \sum_{t=1}^{T-1} \alpha_t \|H_t^{-1/2} m_t\|^2 + \frac{\alpha}{(1 - \beta_1)} \sum_{i=1}^{d} \sum_{j=1}^{T} \frac{\beta_1^{T-j} g_{j,i}^2}{\sqrt{j g_{j,i}^2}}$$

$$\leq \sum_{t=1}^{T-1} \alpha_t \|H_t^{-1/2} m_t\|^2 + \frac{\alpha}{(1 - \beta_1)} \sum_{i=1}^{d} \sum_{j=1}^{T} \frac{\beta_1^{T-j} |g_{j,i}|}{\sqrt{j}}$$

$$\tag{57}$$

where the first inequality is from the Cauchy-Schwarz inequality. The second inequality is due to the fact that $\beta_{1t} \leq \beta_1, \forall t$. The third inequality follows from $\sum_{j=1}^{T} \beta_1^{T-j} \leq 1/(1 - \beta_1)$ and that $1 - \beta_{1j} \leq 1$. The fourth one comes from the fact that $h_{t,i}^2 \geq \frac{t-1}{t} h_{t-1,i}^2$ and that $h_{t,i}^2 \geq g_{t,i}^2$. Therefore

$$\sum_{t=1}^{T} \alpha_t \|H_t^{-1/2} m_t\|^2 \leq \sum_{t=1}^{T} \frac{\alpha}{(1 - \beta_1)} \sum_{i=1}^{d} \sum_{j=1}^{t} \frac{\beta_1^{t-j} |g_{j,i}|}{\sqrt{j}} = \frac{\alpha}{(1 - \beta_1)} \sum_{i=1}^{d} \sum_{t=1}^{T} \sum_{j=1}^{t} \frac{\beta_1^{t-j} |g_{j,i}|}{\sqrt{j}}$$

$$= \frac{\alpha}{(1 - \beta_1)} \sum_{i=1}^{d} \sum_{t=1}^{T} \frac{|g_{t,i}|}{\sqrt{t}} \sum_{j=t}^{T} \beta_1^{j-t} \leq \frac{\alpha}{(1 - \beta_1)^2} \sum_{i=1}^{d} \sum_{t=1}^{T} \frac{|g_{t,i}|}{\sqrt{t}} \tag{58}$$

$$\leq \frac{\alpha}{(1 - \beta_1)^2} \sum_{i=1}^{d} \|g_{1:T,i}\|_2 \sqrt{\sum_{t=1}^{T} \frac{1}{t}} \leq \frac{\alpha \sqrt{1 + \log T}}{(1 - \beta_1)^2} \sum_{i=1}^{d} \|g_{1:T,i}\|_2$$

The second equality is from rearranging the order of summation. The third inequality comes from the fact that $\sum_{j=t}^{T} \beta_1^{j-t} \leq 1/(1 - \beta_1)$. The second last inequality is due to the Cauchy-Schwarz inequality.

**Lemma C.2** *For the parameter settings and conditions assumed in **Theorem C.1**, we have*

$$\sum_{t=1}^{T} \frac{1}{\alpha_t} \left[ \|H_t^{1/2}(x_t - x^*)\|^2 - \|H_t^{1/2}(x_{t+1} - x^*)\|^2 \right] \leq \frac{D_\infty^2}{2\alpha_T} \sum_{i=1}^{d} h_{T,i} \tag{59}$$

*Proof:* By the definition of $L2$ norm, since $\frac{h_{t,i}}{\alpha_t} \geq \frac{h_{t-1,i}}{\alpha_{t-1}}$ by the conditions in the problem (4)

$$
\begin{aligned}
&\sum_{t=1}^{T} \frac{1}{\alpha_t} \left[ \|H_t^{1/2}(x_t - x^*)\|^2 - \|H_t^{1/2}(x_{t+1} - x^*)\|^2 \right] \\
&\leq \frac{1}{\alpha_1} \|H_1^{1/2}(x_1 - x^*)\|^2 + \sum_{t=2}^{T} \left[ \frac{\|H_t^{1/2}(x_t - x^*)\|^2}{\alpha_t} - \frac{\|H_{t-1}^{1/2}(x_t - x^*)\|^2}{\alpha_{t-1}} \right] \\
&= \frac{1}{\alpha_1} \sum_{i=1}^{d} h_{1,i}(x_{1,i} - x_i^*)^2 + \sum_{t=2}^{T} \sum_{i=1}^{d} \left[ \frac{h_{t,i}}{\alpha_t}(x_{t,i} - x_i^*)^2 - \frac{h_{t-1,i}}{\alpha_{t-1}}(x_{t,i} - x_i^*)^2 \right] \qquad (60) \\
&= \frac{1}{\alpha_1} \sum_{i=1}^{d} h_{1,i}(x_{1,i} - x_i^*)^2 + \sum_{t=2}^{T} \sum_{i=1}^{d} \left[ \frac{h_{t,i}}{\alpha_t} - \frac{h_{t-1,i}}{\alpha_{t-1}} \right] (x_{t,i} - x_i^*)^2 \\
&\leq \frac{D_\infty^2}{2\alpha_T} \sum_{i=1}^{d} h_{T,i}
\end{aligned}
$$

where the first inequality is deleting the last negative term in the summation. The last inequality is from the telescopic summation and the bounded diameter that $\|x - x^*\| \leq D_\infty$

## D    EXPERIMENT DETAILS AND MORE EXPERIMENTS

For all the algorithms, we used the default momentum hyper-parameters, that is $\gamma = 0.9$ for SGDM, $(0.9, 0.999)$ for Adam and AMSGrad, and $(\beta_1, c_t) = (0.9, 1)$ for AMX in algorithm 3. The small number $\epsilon$ is set to be 1e-8 to avoid division by zero.

### D.1    SYNTHETIC EXAMPLE

For the synthetic example in section 4, we used $|g_{t,i}| = \frac{1}{\sqrt{t}}|g_{1,i}|, \forall i$, and $|g_{1,i}| = 0.01$ across all the dimensions. The dimension size is set to be $d = 3$, and the step sizes are set to be $\alpha = 0.5$, $\alpha_t = \alpha/\sqrt{t}$ for all the algorithms. The design of step sizes is much larger than what we usually use in real applications, because it makes the increment in the regret bound of AMX much more visible. The increment in the first term of AMX is already zero, and the increment in the second term is very small. If we use a smaller step size $\alpha$, we can barely see any increment in the regret bound of AMX in the figures. The hyper-parameter of AMSGrad is set to be $\beta_2 = 0.999$.

### D.2    IMAGE CLASSIFICATION

For CIFAR-10 and CIFAR-100, the $32 \times 32$ images were zero-padded on the edges with 4 pixels on each margin and randomly cropped and horizontally flipped to generate the input. The input images were also normalized using the dataset mean and standard deviation. For the step sizes on CIFAR-10, we searched over {1e-4, 5e-4, 1e-3, 2e-3, 3e-3, 5e-3, 1e-2, 2e-2} for all the adaptive methods and found that 5e-3 works the best for AMX. The best step sizes for AdaGrad, Adam, AMSGrad were 1e-2, 1e-3, 1e-3 respectively. For SGDM, the best step size was 1e-1. We also grid-searched the weight decay on {1e-1, 5e-2, 1e-2, 5e-3, 1e-3, 5e-4, 1e-4}. On CIFAR-10, 1e-1 weight decay worked the best for Adam and AMSGrad and 5e-2 worked the best for AMX. 5e-4 worked the best for AdaGrad and SGD. On CIFAR-100, all adaptive algotithms worked the best when the weight decay was 1e-1, but SGD still needed the 5e-4 weight decay. The increments in the first term and the second term of the regret bounds are plotted in Figure 5(a) and 5(b). On CIFAR-10, the step sizes of different algorithms were decreased by a factor of 0.1 at the the 100th and 150th epoch. On CIFAR-100, the step sizes were decreased by a factor of 0.2 at the 60th, 120th, and 160th epoch.

### D.3    IMAGE SEGMENTATION

The Deeplab-ASPP model implemented by Kazuto1011 (2016) was used in this task. We followed their settings and reported the mean IoU values averaged over three independent runs. The model

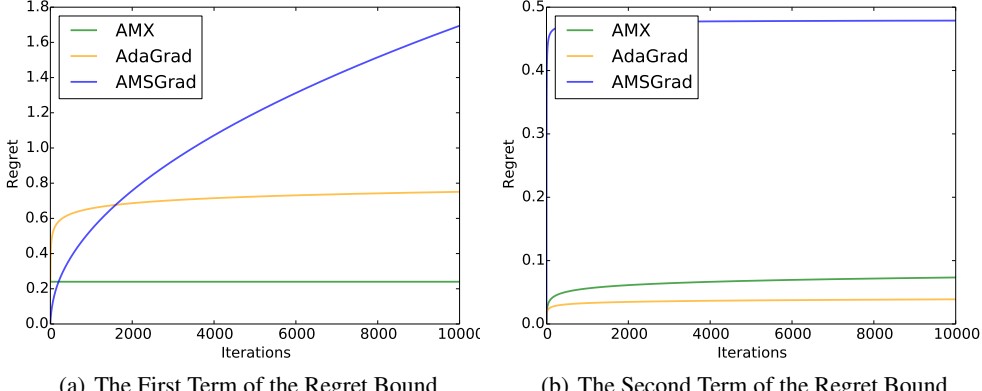

(a) The First Term of the Regret Bound

(b) The Second Term of the Regret Bound

Figure 5: The first and the second term of the regret bound in equation (2) with the diameter $D_{t,\infty}$ replaced by $D_\infty = 2$. As can be observed, the first term of AMX stops increasing after $\tau$, which is the first time step. The other algorithms do not have such a nice property. Therefore, even if the second term of AMX is slightly larger than the second term of AdaGrad, the overall regret of AMX is much smaller. That means AMX converge much faster than AdaGrad and AMSGrad in the example.

was pretrained on the MS-COCO dataset (Lin et al., 2014). We did not use the CRF post-processing technique. We tried the initial step sizes {5e-3, 1e-3, 5e-4, 1e-4, 5e-5, 3e-5, 1e-5, 5e-6} for all the optimizers and we found that 1e-6 worked the best for Adam and AMSGrad. 5e-5 was the best for AMX and 1e-3 was the best for SGD. Similar to the experiments on CIFAR-10, 0.1 weight decay was applied to Adam and AMSGrad. 5e-2 weight decay was applied to AMX and 5e-4 weight decay was applied to SGDM.

### D.4 LANGUAGE MODELING

We trained three layer LSTMs using the instructions provided by Kazuto1011 (2016). Specifically, the LSTMs consisted of 200 embedding size and 1k hidden units. For the dropout probabilities and the batch size, we followed the default values. A 1.2e-6 weight decay was applied to all the algorithms and we tuned over {1e-4, 5e-4, 1e-3, 2e-3, 3e-3, 4e-3, 5e-3, 1e-2, 5e-2} for the initial step sizes. We found that the algorithms were not very sensitive to the initial step sizes, and we reported the results of 2e-3 for Adam and AMSGrad, 3e-3 for AMX, and 1e-2 for AdaGrad, which were the best results among all the possible step sizes. We decayed the step size by a factor of 0.1 at the 300th and 400th epochs.

### D.5 NEURAL MACHINE TRANSLATION

We used the basic implementation of the attentional neural machine translation model by pcyin (2018) and followed their settings. The hidden size of the LSTM was 256 and the embedding size was 256. Label smoothing was set to be 0.1 and the drop out probability was 0.2. We tuned over {5e-2, 1e-2 5e-3, 1e-3, 5e-4, 1e-4} step sizes for all the adaptive optimizers and found that similar to CIFAR-10, 1e-3 worked the best for Adam and AMSGrad and 5e-3, 1e-2 were the best step sizes for AMX and AdaGrad respectively. We averaged the BLEU scores on the IWSLT'14 German to English dataset (Ranzato et al., 2015) over three independent runs and reported them in table 2.

### D.6 EMPIRICAL STUDY OF THE HYPER-PARAMETER

The hyper-parameter $c_t$ plays an important role in our class of AMX algorithms. We conducted an empirical study on some of the designs of $c_t$ and compared the performance of the corresponding algorithms. We compared the designs $c_t = 1, 0.5, 0.1, 1/\sqrt{t}, 1/t$ on CIFAR-10 and reported their top-1 accuracy in Figure 6 and Table 3. All the algorithms were trained using the same settings as Appendix D.2 with the same initial step size 5e-4. As observed, changing the constant $c_t$ to 0.1 and

0.5 did not affect the convergence speed or the final accuracy too much, which meant the performance was not sensitive to the value of $c_t$. However, $c_t = 1/\sqrt{t}$ and $c_t = 1/t$ resulted in slower convergence and worse performance, meaning that these designs were not preferred. These observations also correspond to our claims in section A.5, i.e. despite that these designs make the first term in Theorem 2.1 even smaller, they also make the second term much larger so that the algorithm becomes slower.

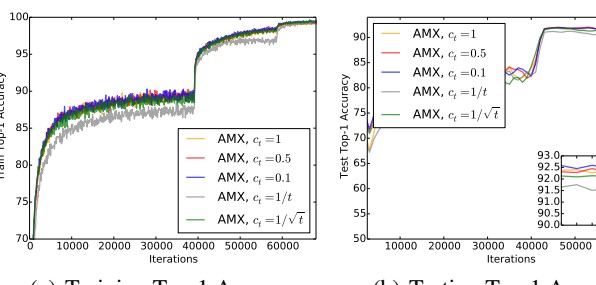

(a) Training Top-1 Accuracy  (b) Testing Top-1 Accuracy

Figure 6: Training and testing Top-1 accuracy curve on CIFAR-10 with different designs of $c_t$

Table 3: Testing Top-1 Accuracy on CIFAR-10 with different designs of $c_t$. The results were averaged over 3 independent runs.

| $c_t$ | ACC. |
|---|---|
| $c_t = 1$ | $92.42 \pm 0.06$ |
| $c_t = 0.5$ | $92.35 \pm 0.08$ |
| $c_t = 0.1$ | $92.46 \pm 0.05$ |
| $c_t = 1/\sqrt{t}$ | $92.15 \pm 0.06$ |
| $c_t = 1/t$ | $91.73 \pm 0.13$ |

