# OpenReview forum: "On the Marginal Regret Bound Minimization of Adaptive Methods"
_ICLR.cc/2021/Conference — Reject_

### Official Review · AnonReviewer4 · 2020-10-19
**Good and interesting paper**

**Rating:** 8
**Confidence:** 3

**Review:**

The paper proposes a new motivation for designing the proximal function of adaptive algorithms. This leads to a new class of adaptive algorithms that achieves marginal optimality and converges faster than existing algorithms in the long term. The claims are substantiated empirically for a wide variety of deep learning tasks as well as empirically. I have no complaints about the comprehensive empirical evaluations and the quality of the theoretical results (if correct). I did not have time to go through all the proofs in the supplementary materials.

I find that the intuition and motivation for the proposed class of methods is generally clear. It seems, however, that this method (as described in Section 3) is a bit ad hoc as the authors are trying to examine a *one-step increment* of the regret. Given from what we know of momentum methods in which the exploitation of longer-term memory may be beneficial, perhaps the authors can comment on whether looking at longer step windows of the regret would be beneficial.

Furthermore, the reviewer is wondering whether there's any benefit to going beyond the use of the *diagonal* proximal function. Would a non-diagonal and more dense proximal term be beneficial?

Corollary 4.1 seems overly conservative because the authors use \tau, the max of the \tau_{m_i}'s to control the overall regret bound. Also, it would be good to avoid the use of the O notation here to show the dependence of the regret (or at least discuss it) on the other parameters of the problem (e.g., the diameter of the problem).

---

> ### Author Response · Authors · 2020-11-13
> **Response to Reviewer 4**
>
> R4: Thank you for your interest in our paper and the great comments! The reason why we consider the one-step increment of the regret is that only the gradient in the next step is available, i.e. we don’t know what is going to happen in the future.  The motivation of using future information is actually explored by AdaGrad, where their design is a hind-sight best solution, i.e. after we know all the gradients. Using a longer step window is somewhere between our method and AdaGrad, but we will definitely think about it in the future.
>
> The usage of full matrix proximal functions may be beneficial, and we include some discussions in Appendix B. However, such a design is computationally expensive as we need to compute the inverse of a full matrix. Therefore, generally, we don’t use full matrix proximal functions. The analysis of full matrix version AMX is one of our future work.
>
> Yes, you are absolutely correct about Corollary 4.1. We state Corollary 4.1 to provide a general idea of why our regret bound is much smaller than $O(\sqrt{T})$ because Theorem 4.1 is heavily loaded with notations. We will definitely consider your advice in our camera-ready version.

---

### Official Review · AnonReviewer3 · 2020-10-25
**Interesting paper on adaptive optimization algorithm**

**Rating:** 5
**Confidence:** 3

**Review:**

##########################################################################

Summary:

The paper provides a new family of adaptive optimization algorithms by designing the proximal function of the adaptive algorithms to minimize marginal regret bound. The paper shows that the regret bound is better than existing algorithms in a sense. The paper also presents simulation study on a variety of domains that compare the proposed algorithms with other commonly used algorithms.


##########################################################################

Reasons for score:

Overall, I vote for rejecting. I appreciate the clear explanation of the motivation behind considering marginal regret bound. However, I have a few concerns outlined in the section below.

##########################################################################

Pros:

1. The paper is easy to follow. The idea of minimizing marginal regret bound is novel and interesting.

2. The paper provides rigorous theoretical proofs for many claims, but also generalizes in a clever way (AMX generalizes Equation 5) to potentially enhance performance.

3. Th paper provides experiments on a variety of tasks in different domains including computer vision and language. The results are very encouraging, as the proposed method is shown to converge either faster or better in most cases compared to other commonly used adaptive algorithms. The paper explains in details how the experiments are done, including relevant parameters and training procedure.


##########################################################################

Cons:

1. The paper shows better bound by introducing a quantity $\tau$ which denotes the largest time steps something happens, and then Corollary 4.1 shows regret bound is roughly $\sqrt{\tau}$. Since $\tau$ is at most T, the paper claims that the bound is better than bounds for other adaptive algorithms, which is simply $\sqrt{T}$. However, I am not convinced that this can only be done for AMX, but not for other adaptive algorithms like Adam or AMSGrad. In other words, what is preventing us from defining $\tau$ in a similar or different way, and then show a bound in terms of $\tau$ for Adam or AMSGrad?

2. The main theoretical insights is that AMX can converge faster than other adaptive algorithms. However, in many experiments AMX did not converge faster than Adam or AMSGrad (e.g. Figure 2). Instead, we often see that the test performance of AMX is better than other adaptive algorithms, i.e., AMX converges to a solution that generalizes better. In this sense, there is a misalignment between theory and practice.

3. The baseline adaptive algorithms (SGDM, AdaGrad, AMSGrad, Adam) are all very commonly used but proposed at least two years ago. It would be better if the paper can show comparison with some more recently proposed adaptive algorithms, like Radam or AdaBound.

---

> ### Author Response · Authors · 2020-11-13
> **Response to Reviewer 3**
>
> R3: Thank you for your interest in our paper and the great comments!
>
> We want to first clarify why other adaptive algorithms like AMSGrad cannot achieve a regret bound that depends on $\tau$. The reason behind it is that the marginal increment of these algorithms is not minimized, i.e. $h_{t,i} /\alpha_t – h_{t-1, i}/\alpha_{t-1} > 0 $ at each time step t.  Therefore the telescopic sum in Theorem 2.1 always reaches $ h_{T,i} /\alpha_T = \sqrt{T} h_{T,i} /\alpha$, which gives an $O(\sqrt{T})$ term. However, our method provides a way of “early stopping”, i.e. the telescopic sum stops to increase after $\tau$, that’s the reason why our method generates a regret that depends on $\tau$ instead of $T$.
>
> You are absolutely correct about AMX not converging faster than AMSGrad in Fig. 2. We have turned over many parameter settings and we are reporting the parameter settings with the highest testing accuracy. The reason why AMX cannot be faster than AMSGrad maybe because the $\tau$ in CIFAR tasks is very large so that the AMX cannot converge very fast in the beginning. However, you can see AMX does catch up with the other methods after the second learning rate decrease, proving our claim that it is at least as fast as the other methods (and it can be potentially even faster)
>
> Thank you for pointing out that we need to compare it with more recent methods. We will definitely do that in future versions of our paper.

---

### Official Review · AnonReviewer1 · 2020-10-28
**recommend to reject**

**Rating:** 4
**Confidence:** 4

**Review:**

This paper proposes a new class of adaptive algorithms inspired by finding an optimal proximal function of adaptive algorithms. They provide a theoretical analysis of the new method showing that it would potentially improve the regret bound of current algorithms. Finally, the proposed method is empirically matched with or superior to other popular algorithms on different tasks.

Overall, the method is novel and backed up by some empirical results. However, I have the following concerns mainly on the theories.

- The theories require a bounded domain, yet it is not justified.
- "*The AMX algorithm is at least as fast as AdaGrad and AMSGrad under the same assumptions*": The regret of AdaGrad is ($\frac{D^2}{\alpha} + \alpha)\sum_{i=1}^d ||g_{1:T,i}||$ under the same setting. While for AMX, the construction of h_t potentially decreases the first term in the regret but also brings an extra log factor to the second term.  In fact,  the second term in Theorem 4.1 is ~$\sum_{i=1}^d(\sqrt{1+\log\tau}||g_{1:\tau,i}||)$.  Overall, since that there is no clue how large $\tau$ is, the regret of AMX may be worse than $\sum_{i=1}^d||g_{1:T,i}||$ when $\tau$ is close to $T$. Indeed,  the bound in Theorem 4.2 is worse than AdaGrad by a log term.
- "*We emphasize that a small $\tau$ is not an assumption on the gradient distribution, but rather a condition that on satisfied.*": There is no citation for this statement nor evidence showing that it is actually happening in the real training process. There is only a specific example to show $\tau$ can be 1, but I'm still not convinced.  Small $\tau$ is the key point to show the potential superiority, it would be better to illustrate how $\tau$ behaves in real training.

---

> ### Author Response · Authors · 2020-11-13
> **Response to Reviewer 1**
>
>
> R1: Thank you for all your valuable comments and great ideas! We want to clarify the following points
> The assumption of a bounded domain is commonly used in the literature of adaptive algorithms, so we simply follow this assumption in our theory. Please check [1] [2] [3] for examples of this assumption.
>
> The regret of most adaptive algorithms have the form of equation (3), including AdaGrad that has $h_{T,i} = \frac{||g_{1:T, i}||}{\sqrt{T}} + \epsilon$, where $\epsilon$ is added to avoid division by zero. We are aware that $\sqrt{T} \frac{||g_{1:T, i}||}{T}$ gives an $O(||g_{1:T, i}||)$  term, however there is an extra $\epsilon$ term. Some recent work has shown that this $\epsilon$ term is actually important in the construction of adaptive methods, for example [4]. That is one reason why we consider a general form of equation (3) as this epsilon cannot be simply ignored. Another reason is that recent Adam variants (AMSGrad, NosAdam) have regrets of the form (3). Their $h_T$ cannot be easily written as a function of ${||g_{1:T, i}||}$, so it would be hard to compare them with AdaGrad. In fact, the regret bound of AMSGrad (and many other algorithms) is $O(\sqrt{T} + \sqrt{1+\log T} {||g_{1:T, i}||})$, which is the same as our worst-case (Thm 4.2). However, in practice, we observe that they are actually at least as fast as AdaGrad, which further shows our claim that the regret is actually determined by the first term in equation (3) (the $O(\sqrt{T})$ part). Our algorithm improves this $O(\sqrt{T})$ term to be $\tilde{O}(\sqrt{\tau})$, that’s why we believe it’s a true improvement.
>
> In summary, you are absolutely correct about the second term in our regret bound (Thm 4.2) might be larger than that of AdaGrad. However, we believe that the dominating term is the first term of size $O(\sqrt{T})$ and hence our algorithm improves over the others. But we will definitely try to prove a better bound in the future.
>
> As we have emphasized in our paper, the regret in the example can be considered as the regret increment in real training processes, i.e., the regret increment will be very small after $\tau$ in real situations, but you are definitely correct that showing where $\tau$ is and how it behaves can help with our argument. We will absolutely consider how to show that in future versions of our paper.
>
> [1]. John Duchi, Elad Hazan, and Yoram Singer. Adaptive subgradient methods for online learning and stochastic optimization.Journal of Machine Learning Research (JMLR), pp. 12:2121–2159, 2011
>
> [2] Haiwen Huang, Chang Wang, and Bin Dong. Nostalgic adam: Weighting more of the past gradients when designing the adaptive learning rate.arXiv preprint arXiv: 1805.07557, 2019.
>
> [3] Liangchen Luo, Yuanhao Xiong, Yan Liu, and Xu Sun.  Adaptive gradient methods with dynamic bound of learning rate.Proceedings of 7th International Conference on Learning Representations,2019
>
> [4] Manzil Zaheer, Sashank Reddi, Devendra Sachan, Satyen Kale, and Sanjiv Kumar. Adaptive methods for nonconvex optimization. Advances in Neural Information Processing Systems 31, 2018

---

### Official Review · AnonReviewer2 · 2020-10-31
**A generalized variant of AdaGrad motivated by a greedy optimization of marginal regret**

**Rating:** 5
**Confidence:** 3

**Review:**

The paper proposes a variant of AdaGrad which optimizes greedily the increase of the regret per trial. More precisely, it optimizes diagonal matrices which adaptively determines the regularizer of the composite mirror descent updates. The experimental results shows that the convergence speed is comparable to other state-of-the-arts, but often it achieves better test performances.

The proposed update makes sense and seems natural. However, the modification seems a bit incremental. The authors pointed out the proposed algorithm achieves potentially better regret bound than typical \sqrt{T} and this is the first algorithm to do so. I am afraid that that is not correct. In the online convex optimization literature, there are bunch of researches on learning on "easy data", which aims to obtain better worst case regret bounds by taking advantage of “easyness” in the data. In fact, there are many papers on that topic (see, e.g., NIPS workshop on Learning faster from easy data II in 2015 and following work).

It is not clear to me why the proposed method achieves better test performances in the experiments. The authors should try to explain why. A reasonable explanation might increase the technical contribution.

---

> ### Author Response · Authors · 2020-11-13
> **Response to Reviewer2**
>
> R2: Thank you for all your valuable comments!
>
> We want to emphasize that one method is the first “adaptive algorithm” that achieves a potentially better regret bound than $O(\sqrt{T})$. We don’t pose any assumptions on the easiness of the data apart from the standard assumptions of adaptive methods, i.e. $\|g_{1:T,i}\| \ll \sqrt{T}$. For examples of adaptive methods that have a $\sqrt{T}$ regret bound, please check the following papers [1][2][3][4]. As far as we are aware, our method is the first adaptive algorithm that has such regret bounds. Could you be more specific about which papers in the workshop you are referring to? We would love to check on them.
>
> The better testing performance of our algorithm is an unpredicted advantage of our novel algorithm. The main purpose of this paper is to show that our algorithm is at least as fast as the other adaptive algorithms (Thm 4.1 and 4.2), and it can be potentially even faster in some settings where $\tau$ is small(e.g. Fig 3 in our paper). But we will definitely try to show why our algorithm has a better testing performance in future versions of our paper.
>
> [1] Sashank J. Reddi,  Stayen Kale,  and Sanjiv Kumar.   On the convergence of adam and beyond.Proceedings of the 6th International Conference on Learning Representations (ICLR), 2018.
>
> [2] Haiwen Huang, Chang Wang, and Bin Dong. Nostalgic adam: Weighting more of the past gradientswhen designing the adaptive learning rate.arXiv preprint arXiv: 1805.07557, 2019.
>
> [3] Liangchen Luo, Yuanhao Xiong, Yan Liu, and Xu Sun.  Adaptive gradient methods with dynamicbound of learning rate.Proceedings of 7th International Conference on Learning Representations, 2019
>
> [4] Wenjie Li, Zhaoyang Zhang, Xinjiang Wang, and Ping Luo. Adax: Adaptive gradient descent with exponential long term memory. arXiv preprint arXiv:2004.09740, 2020.

---

### Official Review · AnonReviewer5 · 2020-11-06
**Interesting idea, some concerns about theoretical comparison to prior work.**

**Rating:** 3
**Confidence:** 4

**Review:**


This paper introduces a new online convex optimization algorithm that operates in via the reduction to online linear optimization in which the regret is bounded by $\sum_{t=1}^T \langle g_t, x_t - u\rangle$ where $g_t$ is the gradient of the t^th loss at $x_t$. The algorithm is based on online mirror descent with non-decreasing quadratic regularizers $x^\top H_t x$, using the update $x_{t+1} = argmin_x \langle g_t, x \rangle + (x-x_t)^\top H_t(x-x_t)$ (or, equivalently using the terminology in the paper, $argmin \langle g_t, x_t\rangle /\sqrt(t) + (x-x_t)^\top H_t(x-x_t)$ where we replace $H_t$ by $H_t/\sqrt{t}$. The analysis is restricted to diagonal $H_t$, for which we can break the regret into a sum of $d$ 1-dimensional problems, so it suffices to do the analysis in the scalar case. The idea is to break out the standard analysis of mirror descent regret as the sum over all t of $D^2(H_t - H_{t-1}) + g_t^2 H_t^{-1}$, where $D$ is the $\ell_\infty$ diameter of the domain, and then choose $H_t$ to minimize each of these terms greedily subject to the non-decreasing condition. A regret bound is provided for this algorithm that achieves worst-case $\sqrt{T}$ regret, but in cases in which the gradients are small, the regret is much better. By employing this strategy on a per-coordinate basis one can obtain an adagrad-esque regret bound.

As far as I can tell, the algorithm is equivalent to the update:
$$
x_{t+1} = argmin_x \langle g_t, x\rangle + (x-x_t)^2 max_{t’ \le t} \sqrt{t’}|g_t’|
$$
Where I remove the composite term and consider 1-d problems for simplicity.

The analysis seems correct here, and the idea is a good approach. However, I am a bit concerned about the quality of the theoretical results obtained. In particular, it is extremely unclear to me that the main regret bound in Theorem 4.1 actually offers any advantage whatsoever over AdaGrad. In the paragraphs following the result, the authors offer the example in which $g_t = O(1/\sqrt{t})$. The authors then observe that in this setting, AdaGrad will obtain $O(\sqrt{\log(T)})$ regret. They then also note that Theorem 4.1 obtains regret $\ll \sqrt{T}$. This is true, but my reading of theorem 4.1 is that it will have $O(\log(T))$ regret, which is *worse* than AdaGrad by a $\sqrt{\log(T)}$ factor.
I am not sure what is happening in Figure 1- my understanding is that this is plotting the analytical regret bound which seems actually smaller for AdaGrad using the provided example. I suspect the learning rates for adagrad are not tuned properly, but perhaps I am missing something.

I have been unable to conceive of any sequence of gradients in which theorem 4.1 actually outperforms AdaGrad, and it is very easy to find sequences in which it does much worse (e.g. simply reverse the sequence of gradients in the provided example). It is, however, plausible to me that in reasonable settings in which the gradients decrease over time one might expect the bound to be within a constant (or maybe a log factor) of AdaGrad’s bound.

Looking at the analysis, I suspect that this is a fundamental issue with the approach of bounding bregman divergences based on the diameter of the domain. Once we commit to this, it is clear that the final regret bound must be at least on the order of $D^2H_T$ by telescoping sum, since $H_T$ is increasing. Further, the $\sum_t g_t^2H_t^{-1}$ term can be lower-bounded by $\sum g_t^2 H_T^{-1}$. Clearly, the minimizing value for $H_T$ here is then $D\sqrt{\sum g_t^2}$ to yield a bound of O(D \sqrt{\sum_t g_t^2}), which is exactly what AdaGrad does.

I am willing to believe that the analysis can be improved here, but I am pretty sure that one cannot due so by bounding the Bregman divergences with the diameter. This leads to a pleasant telescoping sum, but I think AdaGrad may already optimize that style of analysis.

As for the empirical results, these seem a bit more promising so perhaps there is some improved analysis that could be made. There could be a few clarifications here though: is the value for $\alpha_t$ set to $\alpha/\sqrt{t}$ in these experiments, or is it tuned via some other schedule? In analysis of Adam and AMSGrad, the authors typically use the $\alpha/\sqrt{t}$ approach, but in practice I think this is not usually employed. If $\alpha_t$ is set via some other schedule, and momentum is used as well, then it becomes unclear if the $\alpha$ tuning and the momentum is not what is providing the gain over AdaGrad rather than other differences.

I am less expert in evaluating the significance of the final numbers in the empirical study. They do seem relevant, but frankly I feel that the theoretical discussion is currently dragging the paper down quite a bit.

---

> ### Author Response · Authors · 2020-11-13
> **Response to Reviewer 5**
>
> R5: Thank you for your interest in our paper and your valuable feedback. We want to clarify the following.
>
> You are definitely correct about the analysis part. Thank you for being such a careful reader. However, as we mention to another reviewer, the regret bound of AdaGrad also has the form of eqn (3), i.e. $O(\sqrt{T})$, which has $h_{T,i} = \frac{||g_{1:T, i}||}{\sqrt{T}} + \epsilon$, where $\epsilon$ is added to avoid division by zero. We are aware that $\sqrt{T} \frac{||g_{1:T, i}||}{T}$ gives an $O(||g_{1:T, i}||)$ bound, however there is an extra $\epsilon$ term. Some recent work have shown that this $\epsilon$ term is actually important in the construction of adaptive methods, for example [1], so it cannot be simply ignored. Another reason is that recent variants of Adam (AMSGrad, NosAdam etc) have regret bounds of form (3), and their $h_{T,i}$ cannot be easily written as a function of $||g_{1:T, i}||$, so it’s hard to compare them with AdaGrad. In fact, the regret bounds of these algorithm have the form of $O(\sqrt{T} + \sqrt{1+\log T}  {||g_{1:T, i}||})$, which is the same as our worst case (Thm 4.2). However, in practice we observe that they are actually at least as fast as AdaGrad, which further proves our claim that the regret is determined by the first term in equation (3) (the $O(\sqrt{T}) $ part). Our algorithm improves this $O(\sqrt{T})$ term to be $\tilde{O}(\sqrt{\tau})$, that’s why we believe it’s a true improvement.
>
> You are definitely correct that our second term could be larger than that of AdaGrad, and we will try to prove a more promising bound in the future.
>
> In terms of the experiments, we are employing the hyper-parameters that generate the best state-of-the-art performances. As we mention in section 5, we have relegated all the details of hyperparameter tuning to Appendix D. When we use momentum, step size schedules and weight decay, we use them for all the optimization algorithms, so we believe they are not the reasons why our algorithm is better. Please feel free to ask any questions if you have doubts about them.
>
> [1] Manzil Zaheer, Sashank Reddi, Devendra Sachan, Satyen Kale, and Sanjiv Kumar. Adaptive methods for nonconvex optimization. Advances in Neural Information Processing Systems 31, 2018

---

### Decision · Program_Chairs · 2021-01-07
**Final Decision**

**Decision:**

Reject

**Comment:**

The paper presents a new online convex optimization algorithm that uses per-coordinate learning rates. The learning rates are changed over time using information coming from the gradients. A regret upper bound is proved and the algorithm is empirically validated on deep learning experiments.

While the analysis is in principle correct, it does not seem to provide any advantage over the guarantees of similar algorithm, for example the mirror descent version AdaGrad with diagonal matrices. Also, despite the intuition of the authors, the reviewers have found that the approach used in the analsysis is fundamentally bounded to give a worse guarantee than AdaGrad. Overall, the theoretical contribution appears to be not sufficient.

On the empirical side, the experiments failed to convince the majority of the reviewers that the algorithm has a significative gain over similar algorithms.

More generally, this paper suffers from the same problem of many other similar papers: There is a complete disconnect from the theory proven under restrictive assumptions (convexity, bounded domains, no stochasticity) and the experiments (non-convex functions, no projection on bounded domain, stochastic setting). Unfortunately, the deep learning literature is full of such papers, but the community should strive to do better and substantially raise the quality of field. In this view, I strongly suggest to the authors to try to improve the theoretical contribution, for example, trying to prove a convergence guarantee of the gradients to 0, rather than focusing on regret upper bounds. Such analysis would also suggest better ways to design new optimization algorithms better suited to non-convex problems.